# Graphical Resource Allocation with Matching-Induced Utilities

## Abstract

Motivated by real-world applications, we study the fair allocation of graphical resources, where the resources are the vertices in a graph. Upon receiving a set of resources, an agent's utility equals the weight of the maximum matching in the induced subgraph. We care about maximin share (MMS) fairness and envy-freeness up to one item (EF1). Regarding MMS fairness, the problem does not admit a finite approximation ratio for heterogeneous agents. For homogeneous agents, we design constant-approximation polynomial-time algorithms, and also note that significant amount of social welfare is sacrificed inevitably in order to ensure (approximate) MMS fairness. We then consider EF1 allocations whose existence is guaranteed. We show that for homogeneous agents, there is an EF1 allocation that ensures at least a constant fraction of the maximum possible social welfare. However, the social welfare guarantee of EF1 allocations degrades to $1/n$ for heterogeneous agents, where $n$ is the number of agents. Fortunately, for two special yet typical cases, namely binary-weight and two-agent, we are able to design polynomial-time algorithms ensuring a constant fractions of the maximum social welfare.

## 1 Introduction

Resource allocation has been actively studied due to its practical applications [Moulin, 2003; Goldman and Procaccia, 2014; Flanigan *et al.*, 2021]. Traditionally, the utilities are assumed to be additive which means an agent's value for a bundle of resources equals the sum of each single item's marginal utility. But in many real-word problems, the resources have graph structures and thus the agents' utilities are not additive but depend on the structural properties of the received resources. For example, Peer Instruction (PI) has been shown to be an effective learning approach based on a project conducted at Harvard University, and one of the simplest ways to implement PI is to pair the students [Crouch and Mazur, 2001]. Consider the situation when we partition students to advisors, where the advisors will adopt PI for their assigned students. Note that the advisors may hold different perspectives on how to pair the students based on their own experience and expertise, and they want to maximize the efficiency of conducting PI in their own assigned students. How should we assign the students fairly to the advisors? How can we maximize the social welfare among all (approximately) fair assignments? In this work, we take an algorithm design perspective to solve these two questions. Similar pairwise joint work also appears as long-trip coach driver vs co-driver and accountant vs cashier, which is widely investigated in matching theory [Lovász and Plummer, 2009].

The graphical nature of resources has been considered in the literature (see, e.g., [Bouveret *et al.*, 2017; Suksompong, 2019; Bilò *et al.*, 2019; Igarashi and Peters, 2019]). In this line of research, the graph is used to characterize feasible allocations (such as the resources allocated to each agent should be connected), but the agents still have additive utilities over allocated items. With graphical resources, the value of a set of resources does not solely depend on the vertices or the edge weights, but decided

by the combinatorial structure of the subgraph, namely, maximum matching in our problem. Graph structure is also considered in cooperative game theory (i.e., hedonic games) Bogomolnaia and Jackson [2002]; Elkind and Wooldridge [2009]; Aziz *et al.* [2019], but this is not a resource allocation problem and its major concern is how stable coalition structure can be formed.

Our problem also aligns with the research of balanced graph partition [Miyazawa *et al.*, 2021]. Although there are heuristic algorithms in the literature [Kress *et al.*, 2015; Barketau *et al.*, 2015] that partition a graph when the subgraphs are evaluated by maximum matchings, these algorithms do not have theoretical guarantees. Our first fairness criterion is the *maximin share* (MMS) fairness proposed by Budish [2011], which generalizes the max-min objective in Santa Claus problem [Bansal and Sviridenko, 2006]. Informally, the MMS value of an agent is her best guarantee if she is to partition the graph into several subgraphs but receives the worst one. We aim at designing efficient algorithms with provable approximation guarantees. As will be clear later, to achieve (approximate) MMS fairness, a significant amount of social welfare has to be inevitably sacrificed. Our second fairness notion is *envy-freeness* (EF) [Foley, 1967]. In an EF allocation, no agent prefers the allocation of another agent to her own. Since the resources are indivisible, such an allocation barely exists, and recent research in fair division focuses on achieving its relaxations instead. One of the most widely accepted and studied relaxations is *envy-freeness up to one item* (EF1) [Budish, 2011], which requires the envy to be eliminated after removing one item. Lipton *et al.* [2004] proved that an EF1 allocation always exists even with combinatorial valuations.[1] It is noted that an arbitrary EF1 allocation may have low social welfare, and our goal is to compute an EF1 allocation which preserves a large fraction of the maximum social welfare without fairness constraints. The social welfare loss by enforcing the allocations to be EF1 is quantified by *price of EF1* [Bei *et al.*, 2021].

## 1.1 Our Results

We study the fair allocation of graphical resources when the resources are indivisible and correspond to the vertices in the graph, and the agents' valuations are measured by the weight of the maximum matchings in the induced subgraphs. The fairness of an allocation is measured by maximin share (MMS) and envy-free up to one item (EF1). We aim at designing efficient algorithms that compute fair allocations with high social welfare. Our main contributions are summarized as follows.

We first consider homogeneous agents when their valuations are identical. For homogeneous agents, the MMS fairness degenerates to the max-min objective, i.e., partitioning the vertices so that the minimum weight of the maximum matchings in the subgraphs is maximized. It is easy to see that an MMS fair allocation always exists but finding it is NP-hard. We design a polynomial-time $1/8$-approximation algorithm for arbitrary number of agents, and show that when the problem only involves two agents, the approximation ratio can be improved to $2/3$. It is noted that, to ensure any finite approximation of MMS fairness, significant amount of social welfare is inevitably sacrificed. Regarding EF1 fairness, we design a polynomial-time algorithm that computes an EF1 allocation whose social welfare is at least $2/3 + 2/(9n - 3)$ fraction of the maximum social welfare that can be achieved without fairness constraints, where $n$ is the number of agents. Note that when $n = 2$, the approximation ratio is 4/5, and we conjecture that there always exists an EF1 allocation that achieves the maximum social welfare for any number of agents.

We then consider the case of heterogeneous agents. Unfortunately, we show strong impossibility results for the general case. Particularly, for MMS fairness, no algorithm has bounded approximation ratio even if there are two agents with binary weights. For EF1 fairness, no EF1 allocation can ensure better than $1/n$ fraction of the maximum social welfare, but this result does not exclude the possibility of constant approximations for two special cases. In fact, for both two-agent case and binary-weight case, we design polynomial-time algorithms that guarantee $1/3$ fraction of the maximum social welfare. Moreover, for the two-agent case, the approximation ratio is the best possible.

## 1.2 Related Works

Two separate research lines are closely related to our work, namely graph partition and fair division.

---

[1]The algorithm in [Lipton *et al.*, 2004] was originally published in 2004 with a different targeting property. In 2011, Budish [2011] formally proposed the notion of EF1 fairness.

*Graph Partition.* Partitioning graphs into balanced subgraphs has been extensively studied in operations research [Miyazawa *et al.*, 2021] and computer science [Buluç *et al.*, 2016]. There are several popular objectives for evaluating whether a partition is balanced. Among the most prominent ones are the max-min (or min-max) objectives, where the goal is to maximize (or minimize) the total weight of the minimum (or maximum) part. Particularly, the vehicle routing problem (VRP) [Koç *et al.*, 2016], which generalizes the travelling salesperson problem (TSP), is closely related to our work. It asks for an optimal set of routes for a number of vehicles, to visit a set of customers. There are a number of popular variants for the VRP, e.g., the so called heterogeneous vehicle routing problem [Yaman, 2006; Rathinam *et al.*, 2020]. There are many other combinatorial structures studied in graph partitioning problems. For example, in the min-max tree cover (a.k.a. nurse station location) problem, the task is to use trees to cover an edge-weighted graph such that the largest tree is minimized [Khani and Salavatipour, 2014]. This problem also falls under the umbrella of a more general problem, the graph covering problem, where a set of pairwise disjoint subgraphs (called templates) is used to cover a given graph, such as paths [Farbstein and Levin, 2015], cycles [Traub and Tröbst, 2020], and matchings [Kress *et al.*, 2015].

*Fair Division.* Allocating a set of indivisible items among multiple agents is a fundamental problem in the fields of multi-agent systems and computational social choice, and we refer the readers to recent surveys [Amanatidis *et al.*, 2022; Aziz *et al.*, 2022] for more detailed discussion. Envy-freeness (EF) and maximin share fairness (MMS) are two well accepted and extensively studied solution concepts. However, with indivisible items, these requirements are demanding and thus the state-of-the-art research mostly studies their relaxations and approximations. For example, EF1 allocation is studied as a relaxation of EF which always exists [Lipton *et al.*, 2004]. Various constant approximation algorithms for MMS allocations are proposed in [Kurokawa *et al.*, 2018; Garg and Taki, 2021] for additive valuations and in [Barman and Krishnamurthy, 2020; Ghodsi *et al.*, 2018] for subadditive valuations. Our work focuses on indivisible graphical items where agents have combinatorial valuations (neither subadditive nor superadditive) depending on the structural properties. Moreover, all the existing algorithms for non-additive valuations run in polynomial time only if the computation of valuations is assumed to be effortless (i.e., oracles). In contrast, in this work, we aim at designing truly polynomial-time approximation algorithms without valuation oracles.

## 2 Preliminaries

Denote by $G = (V, E)$ an undirected graph without reflexive edges, where $V$ contains all vertices and $E$ contains all the edges. The vertices are the items that are to be allocated to $n$ heterogeneous agents, denoted by $N$. Each agent $i$ has an edge weight function $w_i : E \to \mathbb{R}^+ \cup \{0\}$, which may be different from others'. If $w_i(e) \in \{0, 1\}$ for all $e \in E$, then the weight function is called binary. Let $\mathbf{w} = (w_1, \cdots, w_n)$. A matching $M \subseteq E$ is a set of vertex-disjoint edges, and let $w_i(M) = \sum_{e \in M} w_i(e)$. For any subgraph $G'$, let $V(G')$ and $E(G')$ be the sets of vertices and edges in $G'$, respectively. An allocation $\mathbf{X} = (X_1, \cdots, X_n)$ is a partition of $V$ such that $\cup_{i \in N} X_i = V$ and $X_i \cap X_j = \emptyset$ for $i \neq j$. If $\cup_{i \in N} X_i \subsetneq V$, the allocation is called *partial*. Each agent $i$ has a utility function $u_i : 2^V \to \mathbb{R}^+ \cup \{0\}$, where $u_i(X_i)$ equals the weight of a maximum (weighted) matching in $G[X_i]$. When the agents have identical valuations (i.e., homogeneous agents), we omit the subscript and use $w(\cdot)$ and $u(\cdot)$ to denote all agents' weight and utility functions. A problem instance is denoted by $\mathcal{I} = (G, N)$. When we want to highlight the weight function, $w$ is also included as a parameter, i.e., $\mathcal{I} = (G, N, w)$.

Next we introduce the solution concepts. Our first fairness notion is *maximin share* (MMS) [Budish, 2011]. Letting $\Pi_n(V)$ be the set of all $n$-partitions of $V$, the maximin share of agent $i$ is

$$\mathsf{MMS}_i(\mathcal{I}) = \max_{\mathbf{X} \in \Pi_n(V)} \min_{j \in N} u_i(X_j).$$

We may write $\mathsf{MMS}_i$ for short if $\mathcal{I}$ is clear from the context. Therefore agent $i$ is satisfied regarding MMS fairness if her utility is no smaller than $\mathsf{MMS}_i$.

**Definition 2.1** ($\alpha$-MMS). *For any $\alpha \geq 0$, an allocation $\mathbf{X} = (X_1, \cdots, X_n)$ is called $\alpha$-approximate maximin share ($\alpha$-MMS) fair if for all agents $i \in N$,*

$$u_i(X_i) \geq \alpha \cdot \mathsf{MMS}_i.$$

*The allocation is called MMS fair if $\alpha = 1$.*

The second fairness notion is about *envy-freeness* (EF). An allocation $\mathbf{X}$ is called EF if no agent envies any other agent's bundle, i.e.,

$$u_i(X_i) \geq u_i(X_j) \text{ for all agents } i, j \in N.$$

We can observe that it is very hard to satisfy EF for an arbitrary instance. Consider a simple counter example, where the graph is a triangle and two agents have weight 1 for all edges. Then in every allocation, there is one agent who gets at most one vertex (with utility 0) and the other agent gets at least two vertices (which contains an edge and thus has utility 1). Accordingly, we focus on the *envy-free up to one item* instead [Budish, 2011].

**Definition 2.2** (EF1). *An allocation* $\mathbf{X} = (X_1, \cdots, X_n)$ *is called* envy-free up to 1 item *(EF1) if for any $i$ and $j$, there exists $g \in X_j$ such that $u_i(X_i) \geq u_i(X_j \setminus \{g\})$.*

Besides fairness, we also want the allocation to be efficient. Given an allocation $\mathbf{X} = (X_1, \cdots, X_n)$, the *social welfare* of $\mathbf{X}$ is $\mathsf{sw}(\mathbf{X}) = \sum_{i \in N} u_i(X_i)$. Note that given any instance $\mathcal{I}$, the best possible social welfare of any allocation is the weight of a maximum matching in the graph $G$ by setting the weight of each edge to $\max_{i \in N} w_i(e)$, which is denoted by $\mathsf{sw}^*(\mathcal{I})$. If the instance $\mathcal{I}$ is clear from the context, we also denote $\mathsf{sw}^*(\mathcal{I})$ as $\mathsf{sw}^*$ for short.

## 3 Homogeneous Agents

We start with the case of homogeneous agents when the agents have identical valuations.

### 3.1 MMS Fair Allocations for Homogeneous Agents

With identical valuations, the MMS fairness degenerates to the max-min objective, where the problem is to partition a graph into $n$ subgraphs so that the smallest weight of the maximum matchings in these subgraphs is maximized. It is easy to see that finding such an allocation is NP-hard even when there are two agents and the graph contains a set of disjoint edges, which is essentially a Partition problem. Therefore, we aim at designing polynomial-time approximation algorithms to achieve the MMS fair objective. Without loss of generality, in this section, we assume $w(e) \geq 1$ for all $e \in E$. Since the agents have identical valuations, we omit the subscript in $\mathsf{MMS}_i$ and simply write MMS.

Our main result in this section is as follows.

**Theorem 3.1.** *We can compute a $1/8$-MMS allocation in polynomial time for homogeneous agents.*

Before proving the theorem, we explain the intuition of Algorithm 1. Given an instance $\mathcal{I} = (G, N)$, to ensure the maximum matching in every subset of vertices to be large, we first try to allocate a maximum matching in the original graph. Specifically, we compute a maximum matching in $G$ denoted by $M^* \subseteq E$, and then partition $M^*$ into $n$ bundles $(M_1, \cdots, M_n)$ where $w(M_1) \geq \cdots \geq w(M_n)$ such that $w(M_n)$ is as large as possible. This task is NP-hard and thus we instead use the following simple greedy solution, which we call *greedy partition* of $M^*$.

**Greedy Partition.** Given a matching $M$, partition $M$ into $\Gamma(M) = (M_1, \cdots, M_n)$ as follows.

- Sort and rename the edges in $M$ such that $w(e_1) \geq \cdots \geq w(e_k)$ where $k = |M|$.
- Initially set $M_1 = \cdots = M_n = \emptyset$.
- For $i = 1, \cdots, k$, select $j$ such that $w(M_j) \leq w(M_{j'})$ for all $j'$ and set $M_j = M_j \cup \{e_i\}$.
- Sort and rename $M_1, \cdots, M_n$ so that $w(M_1) \geq \cdots \geq w(M_n)$.

The greedy partition of $M^*$ corresponds to an allocation of vertices where unmatched vertices $V' = V \setminus \cup_{i \in N} V(M_i)$ can be allocated arbitrarily. The good news is that such an allocation achieves MMS fairness when the graph is unweighted, i.e., $w(e) = w(e')$ for all $e, e' \in E$.

**Lemma 3.2.** *If $G$ is unweighted, the greedy partition $(M_1, \cdots, M_n)$ of $M^*$ is an MMS allocation.*

The bad news is that such an allocation does not have any bounded approximation guarantee when the edges have distinct weights. Consider the following example with two agents and the graph is shown in Figure 1 where $\Delta > 1$ is arbitrarily large. Any allocation with bounded approximation ratio

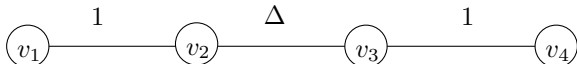

Figure 1: A bad example when greedy partition does not have bounded approximation guarantee of MMS.

of MMS fairness ensures that every agent has value 1, but by partitioning the maximum matching (which contains a single edge with weight $\Delta$) the smaller bundle has value 0. However, if $|M_1| \geq 2$, such an allocation is $1/2$-MMS.

**Lemma 3.3.** *If $|M_1| \geq 2$, $\Gamma(M^*)$ corresponds to an allocation that is $1/2$-MMS fair.*

The tricky case is when $M_1$ contains a single edge $e^*$. To use the approach in Lemma 3.3 to derive $1/2$-MMS fair, we iteratively decrease the weight of $e^*$ and re-compute a maximum matching until $|M_1| \geq 2$. For simplicity, assume all edge weights are powers of 2. This is without much loss of generality which decreases the approximation ratio by at most $1/2$.

**Lemma 3.4.** *Let $\mathcal{I} = (G, N, w)$ and $\mathcal{I}' = (G, N, w')$ be two instances where $\mathcal{I}'$ is obtained from $\mathcal{I}$ by rounding all edge weights down to the nearest power of 2. If $(X_1, \cdots, X_n)$ is an $\alpha$-MMS allocation of $\mathcal{I}'$, then it is an $\alpha/2$-MMS allocation of $\mathcal{I}$.*

We prove Lemmas 3.2, 3.3, and 3.4 in the appendix. Now we are ready to describe Algorithm 1. We first compute a maximum matching $M^*$ and its greedy partition $\Gamma(M^*) = (M_1, \cdots, M_n)$ such that $w(M_1) \geq \cdots \geq w(M_n)$. If $|M_1| \geq 2$, combining Lemmas 3.3 and 3.4, we are safe to output the corresponding partition of vertices so that the approximation ratio is at least $1/4$. If $|M_1| = 1$, we consider two cases. If $w(M_n) \geq 1/2 \cdot w(M_1)$, $w(M_n)$ is still not too small and we can stop the algorithm with a constant approximation ratio. However, if $w(M_n) < 1/2 \cdot w(M_1)$, it means the utility of the smallest bundle is much less than that of the largest bundle. Then we update the edge weights: Let $H$ be the edges with weights no smaller than $w(e_1)$ where $e_1$ is the edge in $M_1$, and decrease their weights to $1/2 \cdot w(e_1)$. By repeating the above procedure, eventually we reach an allocation such that $w(M_n) \geq 1/2 \cdot w(M_1)$ or $|M_1| \geq 2$.

---

**Algorithm 1:** Approximately MMS Fair Allocation Algorithm for $n$ Homogeneous Agents

**Input:** Instance $\mathcal{I} = (G, N)$ with $G = (V, E; w)$.
**Output:** Allocation $\mathbf{X} = (X_1, \cdots, X_n)$.
  1: For all $e \in E$, reset
$$w(e) = 2^{\lfloor \log w(e) \rfloor}.$$
  2: Find a maximum matching $M^*$ in $G$. Denote by $V'$ the set of unmatched vertices.
  3: Find the greedy partition $\Gamma(M^*) = (M_1, \cdots, M_n)$ of $M^*$ such that $w(M_1) \geq \cdots \geq w(M_n)$.
  4: **while** $w(M_1) > 2 \cdot w(M_n)$ and $G$ has different weights **do**
  5:     Let $e_1$ be the edge in $M_1$ and $H = \{e \in E \mid w(e) \geq w(e_1)\}$.
  6:     Let $w(e) = w(e_1)/2$ for all $e \in H$.
  7:     Re-compute a maximum matching $M^*$.
  8:     Re-set $V'$ to be unmatched vertices by $M^*$.
  9:     Re-compute the greedy partition $\Gamma(M^*) = (M_1, \cdots, M_n)$ such that $w(M_1) \geq \cdots \geq w(M_n)$.
 10: **end while**
 11: Set $X_i = V(M_i)$ for $i = 1, \cdots, n-1$.
 12: Set $X_n = V(M_n) \cup V'$.
 13: Return allocation $(X_1, \cdots, X_n)$.

---

We are now ready to prove Theorem 3.1.

*Proof of Theorem 3.1.* First, we show Algorithm 1 is well-defined and runs in polynomial time. Every time when the condition of the **while** loop holds, either the graph has different weights and an allocation is returned or the weights of the heaviest edges are decreased by $1/2^k$ with some $k \geq 1$. Thus the **while** loop is executed $O(\max_{e \in E} \log w(e))$ rounds.

Next we prove the approximation ratio. By Lemma 3.4, we only need to consider the instance where the edge weights are powers of 2 and show the allocation is $1/4$-approximate MMS fair. Denote by

$O = (O_1, \cdots, O_n)$ the optimal solution, where $u(O_1) \geq \cdots \geq u(O_n)$ and $\mathsf{MMS}(\mathcal{I}) = u(O_n)$. The first time when we reach the **while** loop, if $w(M_1) \leq 2 \cdot w(M_n)$,

$$w(M_n) \geq \frac{1}{2} \cdot w(M_1) \geq \frac{1}{2} \cdot u(O_n) = \frac{1}{2} \cdot \mathsf{MMS}(\mathcal{I}),$$

where the second inequality holds because $M^*$ is a maximum matching in $G$. Thus the allocation is 1/2-MMS. If all edges have the same weight, then by Lemma 3.2, the allocation is optimal.

We move into the **while** loop if $w(M_1) > 2 \cdot w(M_n)$ and the edge weights are not identical. Note that $w(M_1) > 2 \cdot w(M_n)$ implies $M_1$ contains a single edge denoted by $e_1$. Otherwise consider the last edge added to $M_1$ in the greedy partition, denoted by $e'$. Then $w(M_1 \setminus \{e'\}) \leq w(M_n)$ and $w(e') \leq w(M_n)$, which implies $w(M_1) \leq 2 \cdot w(M_n)$. After the **while** loop, denote by $\mathcal{I}'$ the instance, by $w'(\cdot)$ the new weights with new utility function $u'(\cdot)$, by $O' = (O_1', \cdots, O_n')$ the new optimal solution and by $M'$ the maximum matching with greedy partition $(M_1', \cdots, M_n')$. Then we have the following claim, which is proved in the appendix.

**Claim 3.5.** *After each **while** loop, one of the following two cases holds true.*

- *Case 1.* $w(e_1) \geq 2 \cdot \mathsf{MMS}(\mathcal{I})$*, then* $\mathsf{MMS}(\mathcal{I}') = \mathsf{MMS}(\mathcal{I})$*;*

- *Case 2.* $w(e_1) < 2 \cdot \mathsf{MMS}(\mathcal{I})$*, then* $2 \cdot \mathsf{MMS}(\mathcal{I}') > \mathsf{MMS}(\mathcal{I})$ *and* $w'(M_1') \leq 2 \cdot w'(M_n')$*.*

By Claim 3.5, the **while** loop will not execute Case 2 or it executes Case 1 for several times and then Case 2 for exactly once. If Case 2 is not executed, then the allocation is 1/2-MMS fair and the analysis is the same with the case when the **while** loop is not executed.

If Case 2 is executed once, then by Claim 3.5,

$$w'(M_n') \geq \frac{1}{2} \cdot w'(M_1') \geq \frac{1}{2} \cdot \mathsf{MMS}(\mathcal{I}') \geq \frac{1}{4} \cdot \mathsf{MMS}(\mathcal{I}).$$

Finally, by Lemma 3.4, the allocation is 1/8-MMS for any instance with arbitrary weights. $\qquad\square$

**Remark.** When $n = 2$, we can improve Algorithm 1 and obtain a better approximation ratio of 2/3. Due to the space limit, we provide the refined algorithm in the appendix.

## 3.2 Efficient and EF1 Allocations for Homogeneous Agents

Recall the example shown in Figure 1. The maximum social welfare is $\mathsf{sw}^* = \Delta$, but any allocation with bounded approximation ratio for MMS fairness has social welfare $2 \ll \Delta$, which means to ensure MMS, we lose significant amount of efficiency. Note that the existence of EF1 allocations is guaranteed by the envy-cycle elimination algorithm designed by Lipton *et al.* [2004]. But the social welfare of the returned allocation does not have any guarantee. In this section, we aim at computing an EF1 allocation that also preserves high social welfare.

**Theorem 3.6.** *For any instance* $\mathcal{I} = (G, N)$*, Algorithm 2 returns an EF1 allocation with social welfare at least* $(2/3 + 2/(9n - 3)) \cdot \mathsf{sw}^*(\mathcal{I})$ *in polynomial time.*

We prove Theorem 3.6 in the appendix, and in the following we briefly discuss the idea of Algorithm 2. We first introduce the *EF1-graph*, inspired by the envy-graph introduced in [Lipton *et al.*, 2004]. Given a (partial) allocation $(X_1, \cdots, X_n)$, we construct the corresponding EF1-graph $\mathcal{G} = (N, \mathcal{E})$, where the nodes are agents (and thus are used interchangeably) and there is a directed edge from $i$ to $j$ if $i$ envies $j$ (or $X_j$) for more than one item,

$$u_i(X_i) < u_i(X_j \setminus \{v\}) \text{ for every } v \in X_j.$$

When the agents have identical utility functions, we have the following simple observation.

**Observation 3.7.** *The EF1-graph is acyclic; The in-degree of the agent with smallest utility is zero.*

Similar with Algorithm 1, in Algorithm 2, we first compute a maximum weighted matching $M^*$ and let the corresponding unmatched vertices be $V'$. If $|M^*| \leq n$, by allocating each edge in $M^*$ to a different agent and $V'$ to one agent who has the smallest utility is EF1, since by removing a vertex from an edge, the remaining subgraph does not have edges any more. If $|M^*| > n$, we find a

**Algorithm 2:** Computing EF1 Allocations with High Social Welfare for $n$ Homogeneous Agents

**Input:** Instance $\mathcal{I} = (G, N)$ with $G = (V, E; w)$.
**Output:** Allocation $\mathbf{X} = (X_1, \cdots, X_n)$.
1: Find a maximum matching $M^*$ in $G$. Denote by $V'$ the set of unmatched vertices by $M^*$.
2: Find the greedy partition $(M_1, \cdots, M_n)$ of edges in $M^*$ such that $w(M_1) \geq \cdots \geq w(M_n)$.
3: Set $X_i = V(M_i)$ for $i = 1, \cdots, n$.
4: **if** $|M^*| \leq n$ **then**
5:   Let $X_n = V(M_n) \cup V'$.
6:   Return $(X_1, \cdots, X_n)$.
7: **end if**
8: Construct the EF1-graph $\mathcal{G} = (N, \mathcal{E})$ based on $(X_1, \cdots, X_n)$.
9: Set $Q$ be the agents with positive in-degree.
10: **for** $i \in Q$ **do**
11:   Let $e_i = (v_{i1}, v_{i2})$ be the last edge added to $M_i$ in the greedy-partition procedure.
12:   $X_i = X_i \setminus \{v_{i1}\}$ and $V' = V' \cup \{v_{i1}\}$.
13: **end for**
14: **for** $v \in V'$ **do**
15:   Let $i = \arg\min_{i \in N} u(X_i)$.
16:   Set $X_i = X_i \cup \{v\}$.
17: **end for**
18: Return $(X_1, \cdots, X_n)$.

greedy-partition $\Gamma(M^*) = (M_1, \cdots, M_n)$ of $M^*$ such that $w(M_1) \geq \cdots \geq w(M_n)$. However, by simply assigning $X_i = V(M_i)$ for every $i$, it may not be EF1, which is illustrated in the appendix.

To overcome this difficulty, we utilize the EF1-graph $\mathcal{G} = (N, \mathcal{E})$ on the partial allocation $(V(M_1), \cdots, V(M_n))$. Let $Q \subseteq N$ be the set of agents who have positive in-degree, i.e., are envied by some agent for more than one item. By Observation 3.7, if $\mathcal{G}$ is nonempty, $Q \neq \emptyset$ and $n \notin Q$. Moreover, since $M_n$ has the smallest weight in the greedy partition $\Gamma(M^*)$, $n$ has an edge to every agent in $Q$. We first consider the partial allocation after the **for** loop in Step 10, which is denoted by $Y = (Y_1, \cdots, Y_n)$. We can prove that $Y$ is EF1, and moreover, it ensures the desired social welfare guarantee. Finally, the remaining steps preserve the EF1ness and can only increase the social welfare of the allocation. The formal analysis is deferred to the appendix.

# 4 Heterogeneous Agents

In this section, we discuss the general case of heterogeneous agents. We first show the negative results for MMS and EF1 allocations, and then focus on the special cases when we are able to obtain positive results. Due to space limit, all the results in this section are proved in the appendix.

## 4.1 Negative Results for MMS and EF1 Allocations

We present the main theorems below whose proofs are in the appendix.

**Theorem 4.1.** *No algorithm has bounded approximation guarantee for MMS fairness, even for the case of two agents with non-identical binary weight functions on the graph.*

**Theorem 4.2.** *No algorithm has better than $1/n$ approximation of social welfare for EF1 fairness for heterogeneous agents.*

Theorem 4.1 is very strong in the sense that it excludes the possibility of designing algorithms with bounded approximation ratio for MMS even for the special cases of two-agent or binary weight functions. However, Theorem 4.2 retains this possibility for EF1, and we design polynomial-time algorithms to compute EF1 allocations that ensure constant fractions of the maximum social welfare for these two cases. In the appendix, we complement Theorem 4.2 with a positive result where we design an algorithm that has $\Omega(1/n^2)$ approximation guarantee of social welfare for the general case.

## 4.2 Binary Weight Functions

---

**Algorithm 3:** Computing EF1 Allocations for $n$ Heterogeneous Agents with Binary Weights

---
**Input:** Instance $\mathcal{I} = (G, N, \mathbf{w})$ with $G = (V, E)$.
**Output:** Allocation $\mathbf{X} = (X_1, \cdots, X_n)$.

1: Initialize $X_i \leftarrow \emptyset, i \in N$. Let $M_i$ be the maximum matching in $G[X_i]$ for agent $i$. Denote by $\mathcal{G}' = (N, \mathcal{E})$ the envy-graph on $\mathbf{X}$.
2: Let $P = V \setminus (X_1 \cup \cdots \cup X_n)$ be the set of unallocated items (called *pool*).
3: Partition agents $i \in N$ into $k$ groups $\mathbf{A}(\mathbf{X}) = (A_1, \cdots, A_k)$ such that agents in the same group have the same value, i.e., $u_i(X_i) = u_j(X_j)$ for $i, j \in A_l$ and $l \in [k]$. Assume $A_l$'s are ordered, i.e., $u_i(X_i) < u_j(X_j)$ for agents $i \in A_{t_1}, j \in A_{t_2}$ and $t_1 < t_2$.
4: Let $t \leftarrow 1$ and $\tau \leftarrow |\mathbf{A}|$.
5: **while** $\{t \leq \tau\}$ **do**
6:     // Case 1. Directly Allocate
7:     **if** there exists an agent $i \in A_t$ such that (1) there is an edge $e$ in $G[P]$ with $w_i(e) = 1$ and (2) allocating the two endpoints $v_1, v_2$ of $e$ to agent $i$ does not break EF1 **then**
8:         $X_i \leftarrow X_i \cup \{v_1, v_2\}, P \leftarrow P \setminus \{v_1, v_2\}$.
9:         Update $u_i(X_i)$ for $i \in N$ and the envy-graph $\mathcal{G}'$.
10:        Update the partition of agents in $\mathbf{A}$.
11:        Reset $t \leftarrow 1$ and $\tau \leftarrow |\mathbf{A}|$.
12:     // Case 2. Exchange and Allocate
13:     **else if** there exists agent $j \in N$ and $i \in A_t$ such that $j$ envies $i$ and there exists a subset with minimum size $V^* \subseteq P$ in graph $G$ such that $u_i(V^*) = u_i(X_i)$ **then**
14:        Let $V^* \subseteq P$ be a set with minimum size such that $u_i(V^*) = u_i(X_i)$.
15:        Let $V_j^* \subseteq X_i$ be a set with minimum size such that $u_j(V_j^*) = u_j(X_j) + 1$.
16:        $P \leftarrow (P \setminus V^*) \cup X_j \cup (X_i \setminus V_j^*)$.
17:        $X_i \leftarrow V^*, X_j \leftarrow V_j^*$.
18:        Update $u_i(X_i)$ for $i \in N$ and the envy-graph $\mathcal{G}'$.
19:        Update the partition of agents in $\mathbf{A}$.
20:        Reset $t \leftarrow 1$ and $\tau \leftarrow |\mathbf{A}|$.
21:     **else**
22:        // Case 3. Skip the Current Agent
23:        $t \leftarrow t + 1$.
24:     **end if**
25: **end while**
26: Execute the envy-cycle elimination procedure on the remaining items $P$.
27: Return the allocation $(X_1, \cdots, X_n)$.

---

We first show that if the agents have binary weight functions, we can compute an EF1 allocation whose social welfare is at least 1/3 fraction of the optimal social welfare. Before introducing our algorithm, we recall the *envy-cycle elimination algorithm* proposed by Lipton *et al.* [2004], which always returns an EF1 allocation. Given a (partial) allocation $(X_1, \cdots, X_n)$, we construct the corresponding *envy graph* $\mathcal{G}' = (N, \mathcal{E})$, where the nodes are agents (and thus are used interchangeably) and there is a directed edge from agent $i$ to agent $j$ if and only if $u_i(X_i) < u_i(X_j)$. The *envy-cycle elimination algorithm* runs as follows. We first find an agent who is not envied by the others, and allocate a new item to her. If there is no such an agent, there must be a cycle in the corresponding envy graph. Then we resolve this cycle by reallocating the bundles: every agent gets the bundle of the agent that she envies in the cycle. We repeat resolving cycles until there is an unenvied agent. The above procedures continue until all the items are allocated. Note that in the execution of the algorithm, the agents' utilities can only increase, and the returned allocation is EF1.

It is not hard to verify that the envy-cycle elimination algorithm does not have any social welfare guarantee. There are several reasons. First, the algorithm does not control which item should be allocated to the unenvied agent so that the agent may receive a set of independent vertices. Second, once an item is allocated it cannot be recalled so that we are not able to revise any bad decision we have made. To increase the social welfare, in each round of our algorithm, we try to allocate an edge (i.e., two items) to the agent $i$ with the smallest value so that the social welfare can increase by 1. However, we need to be very careful by allocating two items which may break the EF1 requirement even if $i$ is not envied by the others. If allocating an edge $e$ to $i$ makes some agent $j$ envy $i$ for more

than one item, we check whether $i$ can maintain her utility by selecting a bundle from unallocated items. If so, we execute *exchange* procedure by asking $j$ to (properly) select a bundle from $X_i$ and $i$ to (properly) select a bundle from unallocated items so that the social welfare is increased by 1. All the items in $X_i$ and the items in $X_j$ that are not selected by $i$ are returned to the algorithm. If not, we try to allocate an edge to the agent with the second smallest value by executing the above procedures, and so on. The description is in Algorithm 3 and we prove the following theorem in the appendix.

**Theorem 4.3.** *For any instance $\mathcal{I} = (G, N)$ where agents have binary weights, Algorithm 3 returns an EF1 allocation with social welfare at least $1/3 \cdot \mathsf{sw}^*(\mathcal{I})$ in polynomial time.*

### 4.3 Two Heterogeneous Agents

We then discuss the case of two agents, and show that Algorithm 4 ensures at least 1/3 fraction of the optimal social welfare. Intuitively, in Algorithm 4, we first check whether there is a single edge $e$ for which some agent $i$ has value at least $1/3 \cdot \mathsf{sw}^*(\mathcal{I})$. If so, allocating $e$ to $i$ already ensures $1/3 \cdot \mathsf{sw}^*(\mathcal{I})$. Moreover, this partial allocation is EF1 since the removal of one item in $e$ results in no edges, and thus we can use the envy-cycle elimination algorithm to allocate the remaining vertices, which returns an EF1 allocation and can only increase the social welfare. Otherwise, we compute a social welfare maximizing allocation $(M_1, M_2)$, i.e., $u_1(M_1) + u_2(M_2) = \mathsf{sw}^*(\mathcal{I})$. Without loss of generality, assume $u_1(M_1) \leq u_2(M_2)$. We temporarily allocate $M_i$ to agent $i$ for $i = 1, 2$. If the allocation is not EF1, since $u_1(M_1) \leq u_2(M_2)$, it can only be the case that agent 1 envies agent 2 but agent 2 does not envy agent 1. Then we move items in agent 2's bundle one by one to agent 1. It can be shown that there must be a time after which the allocation is EF1, and the first time when the allocation becomes EF1, the resulting social welfare must be at least $1/3 \cdot \mathsf{sw}^*(\mathcal{I})$. Formally, we have the following theorem. Interestingly, despite the simplicity of Algorithm 4, we can also show that there is no algorithm that has better than 1/3 approximation.

**Theorem 4.4.** *For any instance $\mathcal{I}$ with two heterogeneous agents, Algorithm 4 returns an EF1 allocation with social welfare at least $1/3 \cdot \mathsf{sw}^*(\mathcal{I})$. Moreover, the approximation of 1/3 is optimal.*

---

**Algorithm 4:** EF1 Allocation with tight social welfare guarantee for 2 Heterogeneous Agents

---

**Input:** Instance $\mathcal{I} = (G, N, \mathbf{w})$ with $G = (V, E)$.
**Output:** Allocation $\mathbf{X} = (X_1, X_2)$.
  1: **if** there is $e \in E$ such that $w_i(e) \geq 1/3 \cdot \mathsf{sw}^*(\mathcal{I})$ for some $i = 1, 2$ **then**
  2:     Assign $e$ to agent $i$ and run envy-cycle elimination algorithm for the vertices.
  3: **else**
  4:     Computing a social welfare maximizing allocation $(M_1, M_2)$. Without loss of generality, assume $u_1(M_1) \leq u_2(M_2)$, and assign $M_i$ to agent $i$ for $i = 1, 2$.
  5:     **while** agent 1 envies agent 2 for more than one item **do**
  6:         Reallocate some item $v \in X_2$ to agent 1, i.e., $X_2 \leftarrow X_2 \setminus \{v\}$ and $X_1 \leftarrow X_1 \cup \{v\}$.
  7:     **end while**
  8: **end if**
  9: Return the allocation $(X_1, \cdots, X_n)$.

---

## 5 Conclusion and Future Directions

In this work, we study the fair (and efficient) allocation of graphical resources when the agents' utilities are determined by the weights of the maximum matchings in the obtained subgraphs. We provide a string of algorithmic results regarding MMS and EF1, but also leave some problems open. For example, for the cases of homogeneous agents and binary valuations, we believe EF1 allocations have better social welfare guarantee. It is also interesting to identify hard instances and study the efficiency limit of EF1 allocations. We can also improve the approximation ratio for the MMS allocation among homogeneous agents. Our work also uncovers many interesting future directions. Firstly, regarding MMS, although we show that there is no bounded multiplicative approximation, it may admit good additive or bi-factor approximations. Secondly, we only focus on the matching-induced utilities in this work, and it is intriguing to consider other combinatorial structures such as independent set, network flow and more. Thirdly, we can extend the framework to the fair allocation of graphical chores when agents have costs to complete the assigned items.

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
