# Appendix

The appendix is organized as follows.

- In Appendices A and B, we provide all the missing proofs from the main body of the work;
- In Appendix C, we design a polynomial-time algorithm to compute an EF1 allocation with at least $1/(4n^2)$ fraction of the maximum social welfare for $n$ heterogeneous agents;
- In Appendix D, we present some interesting results connecting EF1 and Nash social welfare;
- In Appendix E, we show a new algorithm to compute a $2/3$-MMS allocation for two agents.

## A  Missing Proofs of Section 3

### A.1  Proof of Lemma 3.2

*Proof.* Without loss of generality, assume all edges have weight 1. In the greedy partition $(M_1, \cdots, M_n)$ of $M^*$, for any $i \in N$,

$$|M_i| \geq |M_n| = \lfloor \frac{|M^*|}{n} \rfloor.$$

Let $(O_1, \cdots, O_n)$ be an optimal max-min allocation. If $\mathsf{opt} = |O_n| > |M_n|$, then for all $i \in N$,

$$|O_i| \geq \lfloor \frac{|M^*|}{n} \rfloor + 1.$$

Thus

$$\sum_{i \in N} |O_i| \geq n \cdot \lfloor \frac{|M^*|}{n} \rfloor + n > |M^*|,$$

which is a contradiction with $M^*$ being a maximum matching. $\square$

### A.2  Proof of Lemma 3.3

*Proof.* Denote by $O = (O_1, O_2, \cdots, O_n)$ the optimal solution before eliminating any edge, where $u(O_1) \geq u(O_2) \geq \cdots \geq u(O_n)$ and $\mathsf{opt}(\mathcal{I}) = u(O_n)$. Under the maximum matching $M$, consider the greedy partition $(M_1, M_2, \cdots, M_n)$, where $u(M_1) \geq u(M_2) \geq \cdots \geq u(M_n)$. In greedy partition procedure, all edges are sorted in descending order of their weights and each time we select the edge with the largest weight in the remaining edge set and allocate it to the bundle with the least total utility. If $|M_1| \geq 2$, consider the last edge $e$ added to $M_1$, we have $w(M_n) \geq w(e)$, since there exists at least one edge added to $M_n$ before edge $e$ is added to $M_1$. Since in the greedy procedure, edges are added to the bundle with least utility, we have $w(M_n) \geq w(M_1/e)$. Furthermore, we have

$$w(M_n) \geq \frac{1}{2}(w(e) + w(M_1/e)) \geq \frac{1}{2}w(M_1)$$
$$\geq \frac{1}{2n}\sum_{i=1}^{n} w(M_i) \geq \frac{1}{2n}\sum_{i=1}^{n} u(O_i)$$
$$\geq \frac{1}{2}u(O_n),$$

and the lemma holds accordingly. $\square$

### A.3  Proof of Lemma 3.4

*Proof.* Let $\mathcal{I}'' = (G, N, w'')$ be the instance obtained from $\mathcal{I}$ by halving all its edge weights. Let $\mathsf{opt}$, $\mathsf{opt}'$ and $\mathsf{opt}''$ be the optimal values of instance $\mathcal{I}$, $\mathcal{I}'$ and $\mathcal{I}''$, respectively. It is easy to see that

$$\mathsf{opt}'' = \frac{1}{2} \cdot \mathsf{opt}.$$

472     Moreover, the weight of all edges in instance $\mathcal{I}'$ is at least as large as that in instance $\mathcal{I}''$, and thus

$$\mathsf{opt}' \geq \mathsf{opt}'' = \frac{1}{2} \cdot \mathsf{opt}.$$

473     Finally, since $u_i(X_i) \geq \alpha \cdot \mathsf{opt}'$ for all $i \in N$, then

$$u_i(X_i) \geq \frac{\alpha}{2} \cdot \mathsf{opt},$$

474     and thus the lemma holds. $\qquad\square$

## A.4    Proof of Claim 3.5

476     *Proof.* We first consider Case 1. For any $O_i$, if $w(e) < w(e_1)$ for all $e \in E(O_i)$, then $u(O_i)$ does not
477     decrease. If $w(e) \geq w(e_1)$ for some $e \in E(O_i)$, $u(O_i) \geq w(e_1) \geq 2 \cdot \mathsf{opt}(\mathcal{I})$ and after decreasing
478     the weights to $w(e_1)/2$, $u'(O_i) \geq \mathsf{opt}(\mathcal{I})$, implying the existence of an allocation with the minimum
479     utility no smaller than $\mathsf{opt}(\mathcal{I})$, which means $\mathsf{opt}(\mathcal{I}') = \mathsf{opt}(\mathcal{I})$.

480     Second, we consider Case 2 when $w(e_1) < 2 \cdot \mathsf{opt}(\mathcal{I})$. It is straightforward that $2 \cdot \mathsf{opt}(\mathcal{I}') > \mathsf{opt}(\mathcal{I})$
481     since $w(e_1) > \mathsf{opt}(\mathcal{I})$ and after decreasing the weights of some edges to $w(e_1)/2$, $u'(O_i) \geq$
482     $w(e_1)/2 > \mathsf{opt}(\mathcal{I})/2$ (If $e_1 \in O_i$, $u'(O_i) \geq w(e_1)/2$. Otherwise, $u'(O_i) = u(O_i) \geq \mathsf{opt}(\mathcal{I}) >$
483     $\mathsf{opt}(\mathcal{I})/2$ ). Next we show $|M_1'| \geq 2$ which implies $w'(M_1') \leq 2 \cdot w'(M_n')$. For the sake of
484     contradiction, assume $M_1' = \{e_1'\}$. Note that at this moment, $e_1'$ must be an edge with the largest
485     weight in the graph, which means $w'(e_1') = w'(M_1') \geq \cdots \geq w'(M_n')$, and thus

$$w'(M') \leq n \cdot w'(e_1') < n \cdot w'(O_n') \leq \sum_{i \in N} w'(O_i').$$

486     This is a contradiction with $M'$ being a maximum matching in $G$, which completes the proof. $\quad\square$

## A.5    Proof of Theorem 3.6

488     Before proving Theorem 3.6, we first show several technical lemmas. In the following, denote by
489     $Y = (Y_1, \cdots, Y_n)$ the partial allocation after the **for** loop in Step 10 of Algorithm 2.

490     **Lemma A.1.** *$Y$ is EF1.*

491     *Proof.* If $Q = \emptyset$, by definition, the allocation is already EF1. In the following, assume $Q \neq \emptyset$. Note
492     that only the agents in $i \in Q$ has one vertex removed from $V(M_i)$ and for any $i \notin Q$, $Y_i = V(M_i)$.
493     Particularly, $Y_n = V(M_n)$.

494     Fix any $i \in N \setminus \{n\}$. Let $(v_{i1}, v_{i2})$ be the edge selected in Step 11, i.e., the edge with the smallest
495     weight in $M_i$. By the definition of greedy partition,

$$u(V(M_n)) \geq u(V(M_i) \setminus \{v_{i1}, v_{i2}\}). \tag{1}$$

496     We have the following claims.

497     **Claim A.2.** *Agent $n$ does not envy agent $i$ for more than one item in the partial allocation $Y$.*

498     *Proof.* The claim is straightforward if $i \notin Q$ since there is no edge between $n$ and $i$. If $i \in Q$, then
499     $Y_i = V(M_i) \setminus \{v_{i1}\}$ and by Inequality (1),

$$u(V(M_n)) \geq u(V(M_i) \setminus \{v_{i1}, v_{i2}\}) = u(Y_i \setminus \{v_{i2}\}),$$

500     implying $n$ does not envy $Y_i$ for more than one item. $\qquad\square$

501     **Claim A.3.** *Agent $i$ does not envy agent $n$ in $Y$.*

502     *Proof.* If $i \notin Q$, the bundles of agent $i$ and $n$ do not change in the **for** loop in Step 10. Since $M_n$ has
503     the smallest weight in the greedy partition of $M^*$, we have

$$u(Y_i) = u(V(M_i)) \geq u(V(M_n)) = u(Y_n).$$

504     If $i \in Q$, since there is an edge from $n$ to $i$, we have

$$u(Y_i) \geq \min_{v \in V(M_i)} u(V(M_i) \setminus \{v\}) > u(V(M_n)),$$

505     which means $i$ does not envy $n$. $\qquad\square$

506 Combining Claims A.2 and A.3, we have that for any two agents $i$ and $j$,
$$u(Y_i) \geq u(Y_n) \geq u(Y_j \setminus \{v\}) \text{ for some } v \in Y_j,$$
507 which means $i$ does not envy $j$ for more than one item. This completes the proof of Lemma A.1. □

508 To prove the approximation ratio of Algorithm 2, we need the following lemma.

509 **Lemma A.4.** $|M_i| \geq 3$ *for all* $i \in Q$.

510 *Proof.* If the in-degree of agent $i$ is non-zero, then agent $n$ must envy $i$ for more than one item, and
$$u(Y_n) < u(V(M_i) \setminus \{v\}) \text{ for any } v \in V(M_i). \tag{2}$$
511 First, it is easy to see that $|M_i| \neq 1$ since the removal of any node $v$ makes the remaining utility be 0
512 and thus Equation (2) does not hold.

513 Next we show $|M_i| \neq 2$. For the sake of contradiction, assume $M_i = \{e, e'\}$ with $e = (v_1, v_2)$ and
514 $e' = (v_1', v_2')$. Without loss of generality, we further assume $w(e) \geq w(e')$. Then it must be that
515 $w(M_n) \geq w(e)$, otherwise $e'$ cannot be added to $M_i$. Note that since $M^*$ is a maximum weighted
516 matching in $G$, $\{e, e'\}$ must be a maximum weighted matching in $G[M_i]$. If there exist edges in
517 $G[M_i]$ whose weights are greater than $w(e)$, these edges must be adjacent to the same node, denoted
518 by $\bar{v}$; otherwise they can form another matching with weight greater than $w(M_i)$. Thus by removing
519 $\bar{v}$ from $G[M_i]$, the maximum matching in the remaining graph contains at most one edge, and all the
520 remaining edges have weight at most $w(e)$, which means the maximum matching in $G[V(M_i) \setminus \{\bar{v}\}]$
521 brings utility no larger than $w(e)$. Therefore,
$$u(V(M_i) \setminus \{\bar{v}\}) \leq w(e) \leq w(M_n),$$
522 which is a contradiction with Equation (2). Combining the above two cases, we have $|M_i| \geq 3$. □

523 Based on the claims and lemmas presented above, we present the proof of Theorem 3.6 below.

524 *Proof of Theorem 3.6.* Let $(X_1, \cdots, X_n)$ be the allocation returned by Algorithm 2. If the allocation
525 is from Step 6, then it must be EF1. This is because $X_n$ has the smallest value and thus nobody
526 envies $n$ and each of $X_i$ with $1 \leq i \leq n - 1$ contains only two nodes which means the removal of
527 one of them brings utility 0 to any agent. It also achieves the optimal social welfare since all edges in
528 $M^*$ are allocated to some agents.

529 Next we consider the case when the allocation is obtained from Step 18. By Lemma A.1, after the **for**
530 loop in Step 10, the partial allocation is EF1. To show the final allocation to be EF1, it suffices to
531 show that the **for** loop in Step 14 preserves EF1. This is true as in each round, only the bundle with
532 the smallest value can be allocated one more item whose removal makes it smallest again.

533 Finally, we consider the social welfare loss. For each agent $i \in Q$, we observe that at most one node
534 will be removed from $V(M_i)$ in the **for** loop in Step 10 and the **for** loop in Step 14 can only increase
535 $i$'s utility. Since the removed node $v_{i1}$ is from the edge with the smallest weight in $M_i$, by Lemma
536 A.4, we have
$$u(X_i) \geq \frac{2}{3} \cdot u(V(M_i)) \text{ for all } i \in N \setminus \{n\},$$
537 Moreover, for agent $n$ and any $i \neq n$,
$$u(X_n) \geq u(M_i \setminus \{(v_{i1}, v_{i2})\}) \geq \frac{2}{3} \cdot u(V(M_i)). \tag{3}$$
538 Therefore
$$\frac{\sum_{i \in N} u(X_i)}{\mathsf{sw}^*} \geq \frac{\sum_{i \in N \setminus \{n\}} \frac{2}{3} \cdot u(V(M_i)) + u(V(M_n))}{\sum_{i \in N} u(V(M_i))}$$
$$= \frac{2}{3} + \frac{1}{3} \cdot \frac{u(V(M_n))}{\sum_{i \in N} u(V(M_i))}$$
$$\geq \frac{2}{3} + \frac{1}{3} \cdot \frac{u(V(M_n))}{(\frac{3}{2}(n-1) + 1) \cdot u(V(M_n))}$$
$$= \frac{2}{3} + \frac{2}{9n - 3},$$
539 where the second inequality is because of Inequality (3) and we complete the proof of Theorem
540 3.6. □

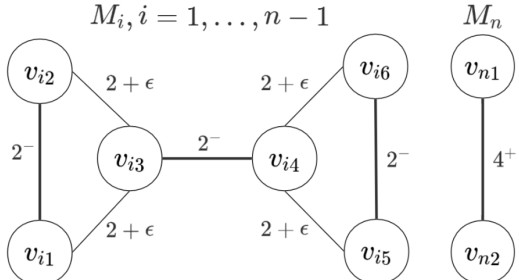

Figure 2: A graph contains $n$ connected components where the first $n - 1$ components are identical as shown by $M_i, i = 1, \cdots, n - 1$, and the last component is a single edge as shown by $M_n$.

**Tight Example.**  We show that the analysis in Theorem 3.6 is asymptotically tight. Consider the example in Figure 2, where $2^-$ means $2 - \epsilon^2$ and $4^+$ means $4 + 3(n - 1)\epsilon^2$. Let $\epsilon > 0$ be a sufficiently small number, say $1/n^2$. The maximum matching $M^*$ contains all the bold edges and $\mathsf{sw}^* = w(M^*) = 6(n - 1) + 4$. By Algorithm 2, the greedy-partition of $M^*$ is $(M_1, \cdots, M_n)$ as shown in Figure 2. However, it is not EF1: for $1 \le i \le n - 1$, by removing any vertex from $M_i$, the maximum matching in the remaining graph has weight at least $4 + \epsilon > 4 + 3(n - 1)\epsilon^2 = w(M_n)$. After the **for** loop in Step 10 in Algorithm 2, for $1 \le i \le n - 1$, one vertex in each $M_i$ is removed and is reallocated to $M_n$ in the **for** loop in Step 14. Thus the remaining social welfare is at most

$$2 \cdot (2 + \epsilon) \cdot (n - 1) + 4 \to \frac{2}{3} \cdot \mathsf{sw}^*.$$

**Remark.**  By Theorem 3.6, if $n = 2$, the approximation ratio is 4/5 and when $n \to \infty$ the approximation ratio is 2/3. Unfortunately, we were not able to prove an upper bound where the optimal social welfare cannot be achieved by any EF1 allocation. We conjecture that there is always an EF1 allocation that achieves the optimal social welfare $\mathsf{sw}^*$.

# B  Missing Proofs of Section 4

## B.1  Proof of Theorem 4.1

*Proof.*  Consider the example as shown in Figure 3. The graph containing four nodes $\{v_1, v_2, v_3, v_4\}$ is allocated to two agents whose valuations (i.e., edge weights) are shown in Figure 3(a) and 3(b) respectively. It can be verified that $\mathsf{MMS}_i = 1$ for both $i = 1, 2$. However, no matter how we allocate the vertices to the agents, one of them receives utility of 0. $\square$

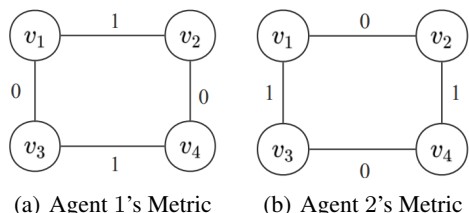

(a) Agent 1's Metric    (b) Agent 2's Metric

Figure 3: A bad example for which no allocation has bounded approximation of MMS fairness.

## B.2  Proof of Theorem 4.2

Similar to Theorem 3.6, we want to find an EF1 allocation which also has high social welfare. Unfortunately, with heterogeneous agents, the fraction of efficiency loss can be as large as $1 - 1/n$.

Note that the optimal social welfare $\mathsf{sw}^*$ is no longer the maximum matching under a single metric, which can be computed by

$$\mathsf{sw}^* = \max_{X \in \Pi_n(V)} \sum_{i \in N} u_i(X_i).$$

*Proof.* Now, we give an instance where, for any $\epsilon > 0$, every EF1 allocation has social welfare at most $(1/n + \epsilon) \cdot \mathsf{sw}^*$. If $\epsilon \geq 1 - 1/n$, it holds trivially since no allocation can have social welfare more than $\mathsf{sw}^*$. In the following, we assume $\epsilon < 1 - 1/n$.

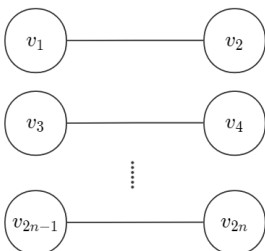

Figure 4: A graph with $n$ disjoint edges is allocated to $n$ agents.

Consider a graph with $n$ disjoint edges, as shown in Figure 4, which is to be allocated to $n$ agents. For each edge, agent 1 has value 1, and the other agents have value $\epsilon$. The maximum social welfare $\mathsf{sw}^* = n$ is achieved by allocating all edges to agent 1. However, to guarantee EF1, at most one edge can be given to agent 1. The maximum welfare of an EF1 allocation is therefore at most $1 + (n-1) \cdot \epsilon$ (each agent receives exactly one edge). The largest ratio is

$$\frac{1 + (n-1) \cdot \epsilon}{n} < \frac{1}{n} + \epsilon,$$

which completes the proof of the Theorem, since $\epsilon$ can be arbitrarily small constant. $\square$

## B.3 Proof of Theorem 4.3

We first prove Theorem 4.3 in Appendix B.3.1 and then show our analysis is tight in Appendix B.3.2.

### B.3.1 The Proof

Before proving Theorem 4.3, we first give several useful lemmas.

**Lemma B.1.** *During the execution of Algorithm 3, the partial allocation maintains EF1.*

*Proof.* During the execution of Algorithm 3, two main cases within the **while** loop in Step 5 change the partial allocation, namely

- Case 1. Directly Allocate;

- Case 2. Exchange and Allocate.

Consider an arbitrary round $t \geq 1$. In Case 1, a single edge is allocated to one agent if and only if such allocation still guarantees EF1. Now, we consider Case 2. If $i \in A_t$ is the agent who is able to pick a subset $V^* \subseteq P$ to maintain his own utility, i.e., $u_i(V^*) = u_i(X_i)$, we show that any other agent does not envy $i$ for more than one item after agent $i$ receives bundle $V^*$. Let $M_i^*$ be the maximum matching of $G[V^*]$ for agent $i$. We first consider agent $j^* \in A_l, l \in [t, \tau]$. Note that replacing $X_i$ by $V^*$ does not change the number of edges in the maximum matching $M_i$ as well as the size of $i$'s bundle $X_i$. Thus, we have

$$u_{j^*}(X_{j^*}) = |M_{j^*}| \geq |M_i| = |M_i^*| = \frac{|V^*|}{2} \geq u_{j^*}(V^*),$$

where the last inequality holds because for binary valuations, the valuation of a bundle for one agent is at most half the size of the bundle. Therefore, agent $j^* \in A_l, l \in [t, \tau]$ does not envy agent $i$ up to

more than one item after $i$ replaces its bundle with $V^*$. Next, we consider agent $j^* \in A_l, l \in [t-1]$. For the sake of contradiction, assume $u_{j^*}(X_{j^*}) < u_{j^*}(V^*)$, which means that there exists at least one edge $e$ such that $w_{j^*}(e) = 1$ as well as a bundle $V_{j^*} \subseteq P$ such that $u_{j^*}(V_{j^*}) = u_{j^*}(X_{j^*})$. Therefore, in the $l$th round of the **while** loop, a single edge $e$ with weight 1 is added to agent $j^*$ if it does not break EF1. Otherwise, there exists an agent $j'$ who envies agent $j^*$ before adding edge $e$. In such case, Algorithm 3 will execute the bundle-exchanging procedure in Step 13-17 in $l < t$th round of the **while** loop, which is a contradiction with the $t$th round of the **while** loop being executed. We complete the proof of Lemma B.1. □

For any graph $G = (V, E)$ and $n$ different binary valuations $v_i(\cdot)$ on $G$, we call a matching $M$ social welfare maximizing if $M$ is a maximum matching on the graph $G' = (V, E')$ where for any $e \in E$ if and only there exists $i$ such that $v_i(e) = 1$. Let $M^*$ denote the social welfare maximizing matching on the input graph $G$ of Algorithm 3. Let $V_R$ be the set of unallocated items after we move out of the **while** loop in Step 25, and $M_R$ be the social welfare maximizing matching on the induced subgraph of $V_R$. Let $V_L$ be the set of allocated items after Step 25 and $M_L$ be the welfare maximizing matching on $V_L$. Actually, $|M_L|$ is the social welfare that Algorithm 3 produces after Step 25. Then we have the following.

**Lemma B.2.** $|M_L| \geq |M_R|$.

*Proof.* We note that when Algorithm 3 moves out of the **while** loop in Step 5, any agent $i$ values the unallocated items no more than its own bundle, i.e., $u_i(X_i) \geq u_i(V_R)$. Then we have $|M_L| = \sum\limits_{i=1}^{n} u_i(X_i) \geq \sum\limits_{i=1}^{n} u_i(V_R) \geq |M_R|$, which completes the proof. □

Based on the claims and lemmas presented above, we are ready to prove Theorem 4.3

*Proof of Theorem 4.3.* Let $M_m$ be a welfare maximizing matching on the bipartite graph induced by $V_L$ and $V_R$. i.e., finding as many disjoint edges $e_{ij}$ as possible such that $v_i \in V_R$ and $v_j \in V_L$. Observe that the maximum number of vertices within $V_L$ equals half the number of the edges in the maximum matching $M_L$, i.e., $|V_L| = 2|M_L|$. Therefore, the size of $M_m$ is at most $2|M_L|$ (each vertex $v_1 \in V_L$ combined with another vertex $v_2 \in V_R$ to form a matching). Therefore, we have $|M^*| \leq 2|M_L| + |M_R|$. Furthermore, we have

$$\frac{u(M_L)}{\mathsf{sw}^*} = \frac{|M_L|}{|M^*|} \geq \frac{|M_L|}{2|M_L| + |M_R|}$$
$$\geq \frac{|M_L|}{3|M_L|} \geq \frac{1}{3},$$

where the second inequality holds because $|M_L| \geq |M_R|$ proved in Lemma B.2. Since Step 26 can only increase the social welfare, we have proved the social welfare guarantee.

It remains to see the running time of the algorithm. In each iteration of the **while** loop in Step 5, the utility of exact one agent increases by 1. Since the maximum possible welfare is bounded by $O(|V|^2)$, the **while** loop will execute for at most $O(|V|^2)$ times. The envy-cycle elimination procedure in Step 26 will execute at most $O(|V|)$ times. Thus, Algorithm 3 runs in $O(|V|^2 + |V|) = O(|V|^2)$ time.

The proof of Theorem 4.3 is completed. □

### B.3.2 Tight Example

We show that the analysis in Theorem 4.3 is asymptotically tight. Consider the example as shown in Figure 5. Let $k > 4$ be a constant. Denote by $\alpha_{ij}, i \in [k], j \in [2]$ the edge between node $v_{ij}$ and node $v'_{ij}, \beta_i, i \in [k]$ the edge between node $v_{i1}$ and $v_{i2}, \gamma_i, i \in [k-1]$ the edge between node $a_{i1}$ and $a_{i2}$. Let $\theta_{i1}, \theta_{i2}, \theta_{i3}, \theta_{i4}, i \in [k]$ be the edge between node $a_{i1}$ and node $v_{11}, a_{i1}$ and node $v_{12}$, $a_{i1}$ and node $v_{21}, a_{i1}$ and node $v_{22}$, respectively. Obviously, allocating all the nodes to agent 2 and allocating nothing to agent 1 result in the optimal social welfare, i.e.,

$$\mathsf{sw}^* = u_2(V) = \sum_{i \in [k], j \in [2]} w_2(\alpha_{ij}) + \sum_{i \in [k-1]} w_2(\gamma_i) = 3k - 1. \tag{4}$$

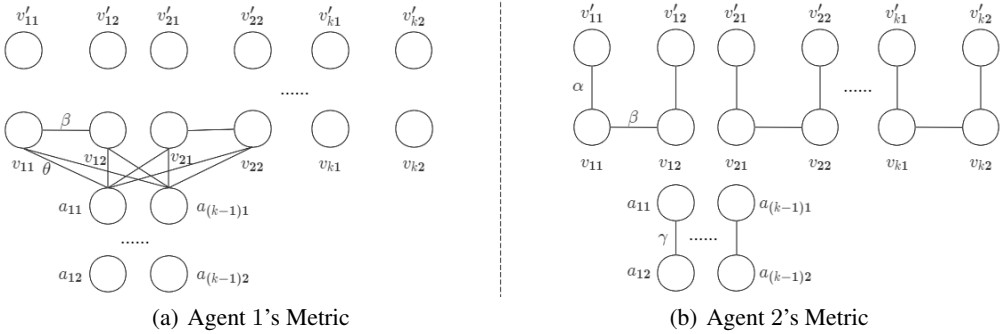

(a) Agent 1's Metric          (b) Agent 2's Metric

Figure 5: The graph is partitioned among two agents with binary valuations.

The corresponding maximum matching $M^*$ contains $2k$ edges $\alpha_{ij}, i \in [k], j \in [2]$ and $k-1$ edges $\gamma_i, i \in [k-1]$. Now, we consider the worst case achieved by Algorithm 3 running on this example, which results in a total utility of $k+3$. In the first two rounds of the **while** loop in Step 5, each agent picks exactly one of the two edges $\beta_1$ and $\beta_2$ (w.l.o.g. agent 1 picks $\beta_1$ and agent 2 picks $\beta_2$). Following that agent 2 picks all the remaining edges $\beta_i, i \in [3, k]$ and arbitrary two edges $\gamma_i, i \in [k-1]$ (w.l.o.g. $\gamma_1$ and $\gamma_2$). We then move out of the **while** loop since (1) for agent 1, $u_1(P) = 0$; (2) for agent 2, $u_2(P) < u_2(X_2)$ and allocating any other edge $\gamma_i, i \in [3, k-1]$ to it will break EF1. Thus, we execute the envy-cycle elimination procedure on the remaining items, i.e., allocating all the remaining vertices in $P$ to agent 1 with the EF1 allocation being completed. For agent 2, the maximum matching in $G[X_2]$ containing edges $\beta_i, i \in [2, k], \gamma_i, i \in [2]$. We thus have $u_2(X_2) = k - 1 + 2 = k + 1$. For agent 1, the maximum matching in $G[X_1]$ containing edges $a_{11}$ and $a_{12}$. Therefore, $u_1(X_1) = 2$. The total social welfare is $u_1(X_1) + u_2(X_2) = k + 1 + 2 = k + 3$. Thus

$$\lim_{k \to +\infty} \frac{k+3}{3k-1} = \frac{1}{3},$$

which completes the proof of the theorem.

### B.4 Proof of Theorem 4.4

We first prove Theorem 4.4 in Appendix B.4.1 and then show that the approximation ratio guarantee is tight, i.e., no algorithm is better than $1/3$-approximate in Appendix B.4.2.

### B.4.1 The Proof

*Proof.* Denote $(M_1, M_2)$ as a social welfare maximizing allocation. Consider the following two cases:

- Case 1: $\exists e \in E$ such that $w_i(e) \geq \frac{1}{3} \cdot \mathsf{sw}^*, i \in \{1, 2\}$;

- Case 2: $\forall e \in E, w_i(e) < \frac{1}{3} \cdot \mathsf{sw}^*, i \in \{1, 2\}$.

For Case 1, giving edge $e$ to agent $i$ and running the envy-cycle elimination procedure on remaining vertices can find an EF1 allocation, which, at the same time, guarantees the total utility no less than $1/3$ of the maximum possible social welfare.

Next, we consider Case 2. There are two subcases.

- Subcase 1: $u_i(M_i) \geq \frac{1}{3} \cdot \mathsf{sw}^*$ for all $i \in \{1, 2\}$;

- Subcase 2: $\exists i \in \{1, 2\}$ such that $u_i(M_i) < \frac{1}{3} \cdot \mathsf{sw}^*$.

For Subcase 1, if such allocation guarantees EF1, the theorem holds. Otherwise, agent 1 envies agent 2 since we assume $u_1(M_1) \leq u_2(M_2)$. We then reallocate the item $v \in X_2$ to agent 1 one by one

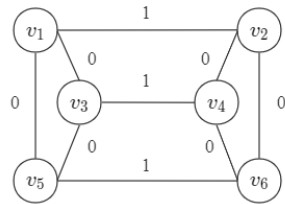

(a) Agent 1's weight for the graph

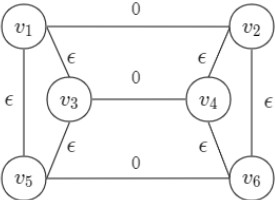

(b) Agent 2's weight for the graph

Figure 6: An example where any EF1 allocation guarantees at most $(1/3 + 2\epsilon)$ of the maximum social welfare.

until such allocation guarantees EF1. The total utility is $u_1(X_1) + u_2(X_2) \geq u_1(M_1) \geq (1/3)\mathsf{sw}^*$. Therefore, we complete the proof for this subcase.

Consider Subcase 2. Without loss of generality, assume $u_1(M_1) < (1/3)\mathsf{sw}^*$. If allocation $(M_1, M_2)$ guarantees EF1, the theorem is proved. Otherwise, by the assumption that $u_1(M_1) \leq u_2(M_2)$, agent 1 envies agent 2 more than one item. By $u_1(M_1) < (1/3)\mathsf{sw}^*$, we have $u_2(M_2) > (2/3)\mathsf{sw}^*$. Now, we consider to remove items from agent 2's bundle to agent 1's bundle. First, sorting the edges within $M_2$ by decreasing order according to their valuation to agent 2. In each iteration, we pick an edge within agent 2's bundle with largest weight and give one endpoint to agent 1. If the allocation still admits EF1, we give another endpoint to agent 1 and pick another edge with largest weight in agent 2's remaining bundle. Repeat above procedure until agent 1 envies agent 2 up to exact one item. When Algorithm 4 completed, at most one edge $e$ within $M_2$ is destroyed, i.e., one endpoint of $e$ is allocated to agent 1 and another endpoint still remains in $X_2$. If $e$ is the edge with largest weight in $M_2$, we have $u_2(X_2) \geq u_2(M_2 \setminus \{e\}) > (1/3)\mathsf{sw}^*$, where the last inequality holds because $u_2(M_2) > (2/3)\mathsf{sw}^*$ and $w_2(e) < (1/3)\mathsf{sw}^*$. We thus complete the proof of the theorem. Otherwise, we next show that $u_2(X_2) \geq (1/3)u_2(M_2)$. Denote by $X_2'$ be the set of items given to agent 1. We have

$$w_2(e) \leq u_2(X_2') \leq u_2(X_2), \tag{5}$$

where the first inequality holds because at least one edge within $M_2$ with larger weight is allocated to agent 1 before and the second inequality holds since otherwise agent 2 will envy agent 1. We thus derive

$$u_2(X_2) \geq \frac{1}{3}(u_2(X_2) + u_2(X_2') + w_2(e)) \geq \frac{1}{3}u_2(M_2). \tag{6}$$

Furthermore

$$
\begin{aligned}
u_1(X_1) + u_2(X_2) &\geq u_1(M_1) + \frac{1}{3}u_2(M_2) \\
&\geq \mathsf{sw}^* - u_2(M_2) + \frac{1}{3}u_2(M_2) \\
&= \mathsf{sw}^* - \frac{2}{3}u_2(M_2) \geq \frac{1}{3}\mathsf{sw}^*,
\end{aligned}
\tag{7}
$$

where the last inequality holds because $u_2(M_2) \leq \mathsf{sw}^*$. Since in each iteration, at most one item is removed from agent 2 to agent 1, Algorithm 4 runs in $poly(|V|)$ time. We complete the proof of the theorem.

### B.4.2 Tight Example

We next show the approximation of $1/3$ is optimal. Consider the example in Fig. 6(a) and Fig. 6(b). It is not hard to verify that the maximum social welfare without fairness constraint is $\mathsf{sw}^* = 3$ by allocating all the items to agent 1. However, for any allocation where agent 1 has utility no smaller than 2, the allocation is not EF1 to agent 2 since agent 2 always has utility 0 in such allocations. Therefore, the maximum social welfare generated by EF1 allocations is no greater than $1 + 2\epsilon$. Thus

$$\lim_{x \to 0} \frac{1 + 2\epsilon}{3} = \frac{1}{3}, \tag{8}$$

which means the approximation ratio of $1/3$ is optimal. □

# C    EF1 Allocation with Bounded Social Welfare Guarantee

Now we are ready to present an algorithm to find an EF1 allocation with bounded social welfare guarantee in polynomial time.

**Theorem C.1.** *For any instance $\mathcal{I} = (G, N)$, Algorithm 5 returns an EF1 allocation with social welfare at least $1/(4n^2) \cdot \mathsf{sw}^*(\mathcal{I})$ in polynomial time.*

---

**Algorithm 5:** Computing EF1 Allocations for $n$ Heterogeneous Agents with Distinct Weights

---
**Input:** Instance $\mathcal{I} = (G, N, w)$ with $G = (V, E)$.
**Output:** Allocation $\mathbf{X} = (X_1, \cdots, X_n)$.
 1: Initialize $X_i \leftarrow \emptyset, i \in N$. Let $M_i$ be the maximum matching in $G[X_i]$ for agent $i$. Denote by $\mathcal{G}' = (N, \mathcal{E})$ the envy-graph on $\mathbf{X}$.
 2: Let $P = V \setminus (X_1 \cup \cdots \cup X_n)$ be the set of unallocated items (called *pool*).
 3: Denote $H$ as the set of agents who are not envied by any other agents. Initialize $H \leftarrow N$.
 4: Let agent $i^*$ determine a maximum matching $M_{i^*}$ in graph $G$. Denote by $R$ the set of the remaining edges within the maximum matching. Initialize $R \leftarrow M_{i^*}$.
 5: Sort the edges $e \in M_{i^*}$ by non-increasing order according to their weight to agent $i^*$.
 6: Let agent $i^*$ pick one edge with largest weight $w_{i^*}(e)$ (with ties broken arbitrarily).
 7: **while** $\{R \neq \emptyset\}$ **do**
 8:     Select one agent $i \in H$.
 9:     **if** $\{P_i = \emptyset\}$ **then**
10:         Select one edge $e \in R$ with largest weight to agent $i^*$. Give one endpoint $v_1$ of $e$ to agent $i$ and put another endpoint $v_2$ in the corresponding pool $P_i$, i.e., $R \leftarrow R \setminus \{e\}$, $X_i \leftarrow X_i \cup \{v_1\}, P_i \leftarrow P_i \cup \{v_2\}, P \leftarrow P \setminus \{v_1, v_2\}$.
11:         Update the envy-graph $\mathcal{G}'$ and set $H$.
12:     **else**
13:         Give the node $v \in P_i$ to agent $i$, i.e., $P_i \leftarrow \emptyset, X_i \leftarrow X_i \cup \{v\}$.
14:         Update the envy-graph $\mathcal{G}'$ and set $H$.
15:     **end if**
16: **end while**
17: Return all the vertices within $P_i$ to the pool $P$, i.e., $P \leftarrow P \bigcup_{i \in N} P_i$.
18: Execute the envy-cycle elimination procedure running on the remaining items $P$.
19: Return the allocation $(X_1, \cdots, X_n)$.

---

Without loss of generality, we assume $i^*$ to be the agent who has the maximum value of $u_i(V), i \in N$. Denote by $S = (S_1, \cdots, S_n)$ the partial allocation when we first move out of the **while** loop in Step 7. Before presenting the proof of Theorem C.1, we first present a useful lemma.

**Lemma C.2.** $\sum\limits_{i \in N} u_{i^*}(X_i) \geq \frac{1}{2}u_{i^*}(V)$.

*Proof.* During the execution of Algorithm 5, there is at most one node $v_i$ in pool $P_i, i \in [n]$. Consider $P_i$ when we first move out of the **while** loop in Step 7. If $P_i \neq \emptyset$, w.l.o.g. suppose $v_i \in P_i$ is one endpoint of edge $e_i \in M_{i^*}$. Let $N_1$ be the set of agents such that $P_i \neq \emptyset$. Since the edges are picked by non-increasing order of their weight to agent $i^*$, we have $u_{i^*}(S_i) \geq w_{i^*}(e_i), i \in N_1$. Furthermore, we have

$$2u_{i^*}(S_i) \geq u_{i^*}(S_i) + w_{i^*}(e_i) \geq u_{i^*}(S_i \cup \{v_i\}).$$

706 Thus, $u_{i^*}(S_i) \geq \frac{1}{2} u_{i^*}(S_i \cup \{v_i\}), i \in N_1$. We have

$$
\begin{aligned}
\sum_{i \in N} u_{i^*}(X_i) &\geq \sum_{i \in N} u_{i^*}(S_i) \\
&\geq \frac{1}{2} \sum_{i \in N_1} u_{i^*}(S_i \cup \{v_i\}) + \sum_{i \in N \setminus N_1} u_{i^*}(S_i) \\
&\geq \frac{1}{2} \sum_{i \in N_1} u_{i^*}(S_i \cup \{v_i\}) + \frac{1}{2} \sum_{i \in N \setminus N_1} u_{i^*}(S_i) \\
&= \frac{1}{2} \sum_{i \in N} u_{i^*}(V),
\end{aligned}
$$

707 where the last equality holds because the nodes within the remaining pool $P$ after we move out of the
708 **while** loop in Step 7 do not have any effect on the maximum matching $M_{i^*}$. We complete the proof
709 of Lemma C.2. $\qquad\square$

710 Now we are ready to prove Theorem C.1.

711 *Proof of Theorem C.1.* Let $N_e$ be the set of agents that agent $i^*$ envies. Since Algorithm 5 admits
712 EF1, there exists one node $v \in X_i, i \in N_e$ such that $u_{i^*}(X_{i^*}) \geq u_{i^*}(X_i \setminus \{v\}), i \in N_e$. For
713 any agent $i \in N_e$, w.l.o.g. assume $v_1$ is the node such that $u_{i^*}(X_{i^*}) \geq u_{i^*}(X_i \setminus \{v_1\})$ and
714 $v_1, v_2 \in S_i$ are two endpoints of the edge $e \in M_{i^*}$. Since agent $i^*$ first picks the edge with the
715 largest weight to itself, we have $u_{i^*}(X_{i^*}) \geq w_{i^*}(e) = u_{i^*}(\{v_1, v_2\})$. By the definition of EF1,
716 $u_{i^*}(X_{i^*}) \geq u_{i^*}(X_i \setminus \{v_1\}) \geq u_{i^*}(X_i \setminus \{v_1, v_2\})$ holds. Thus, we have $u_{i^*}(X_{i^*}) \geq \frac{1}{2}(u_{i^*}(X_i \setminus$
717 $\{v_1, v_2\}) + u_{i^*}(\{v_1, v_2\})) = \frac{1}{2} u_{i^*}(X_i)$. Let $(X_1^*, \cdots, X_n^*)$ be the welfare maximization allocation,
718 where $X_i^*, i \in N$ is the set of vertices allocated to agent $i$. Therefore

$$
\begin{aligned}
n \cdot u_{i^*}(X_{i^*}) &\geq \frac{1}{2} \sum_{i \in N} u_{i^*}(X_i) \geq \frac{1}{2} \cdot \frac{1}{2} u_{i^*}(V) \\
&\geq \frac{1}{4} \cdot \frac{1}{n} \sum_{i \in N} u_i(V) \geq \frac{1}{4n} \sum_{i \in N} u_i(X_i^*) \\
&= \frac{1}{4n} \mathsf{sw}^*(\mathcal{I}),
\end{aligned}
$$

719 where the second inequality follows by Lemma C.2 and the third inequality holds because of the
720 assumption that $i^*$ is the agent with the largest value of $u_i(V), i \in N$. Since in each iteration one
721 node is allocated to an agent, the time complexity of Algorithm 5 is at most $O(|V|^2)$, completing the
722 proof of Theorem C.1. $\qquad\square$

# D Nash Social Welfare and EF1 Allocations

724 Before introducing Nash social welfare, We first generalize the definition of EF1 to envy-free up to $k$
725 items and its approximations. For $\alpha \geq 0$, allocation $(X_1, \cdots, X_n)$ is $\alpha$-approximate envy-free up to
726 $k$ items ($\alpha$-EFk) if for any $i$ and $j$, there exists $S = \{g_1, \cdots, g_k\} \subseteq X_j$ such that

$$
u_i(X_i) \geq \alpha \cdot u_i(X_j \setminus S).
$$

727 Now, we are ready to give several results concerning nash social welfare, which is (informally) the
728 product of all agents' utilities. It is proved in [Caragiannis *et al.*, 2019] that under additive valuations,
729 EF1 and Pareto Optimality (PO) are compatible and an allocation that maximizes Nash social welfare
730 is always simultaneously EF1 and PO. Here, an allocation is PO if there is no alternative allocation
731 that makes an agent better off without making anyone worse off. Recently, Wu *et al.* [2021] showed
732 that with subadditive valuations, a Nash social welfare maximizer is still PO but only 1/4-EF1. In
733 our results, we observe that when the valuations are measured by maximum matchings, Nash social
734 welfare maximizer is PO and EF2 for any matching valuations. However, the Nash social welfare
735 maximizer does not have any bounded approximation guarantee on EF1.

 **Proposition D.1.** *An allocation that maximizes Nash social welfare does not have bounded approxi-*
*mation ratio on EF1.*

*Proof.* Consider the example shown in Figure 7, where the two agents have different metrics as
shown in Figure 7(a) and Figure 7(b) respectively. Let $M$ be a sufficiently large number. The unique
allocation that maximizes the Nash social welfare is to assign $X_1 = \{v_3, v_4, v_5, v_6\}$ to agent 1 and
$X_2 = \{v_1, v_2\}$ to agent 2. However, for any $v \in X_1$,

$$u_2(X_1 \setminus \{v\}) = M \gg 1 = u_2(X_2).$$

When $M$ goes to infinity, the allocation does not have any bounded approximation ratio. $\qquad\square$

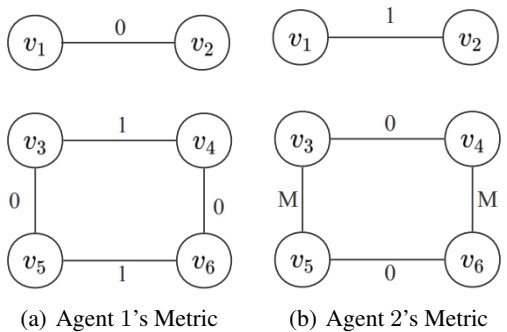

(a) Agent 1's Metric      (b) Agent 2's Metric

Figure 7: The graph containing a single edge and a square is allocated to two agents.

Note that even with identical valuations, the Nash social welfare maximizing allocation does not
guarantee EF1. Consider the example shown in Figure 8, where $\epsilon$ is arbitrarily small. To maximize the
Nash social welfare, one possible allocation is that $X_1 = \{v_1, v_2\}$ and $X_2 = \{v_3, v_4, v_5, v_6, v_7, v_8\}$.
However, this allocation does not guarantee EF1 for agent 1 since the removal of any vertex in $X_2$
still admits a matching with weight at least 8.5, which is greater than $u_1(X_1) = 8 + \epsilon$.

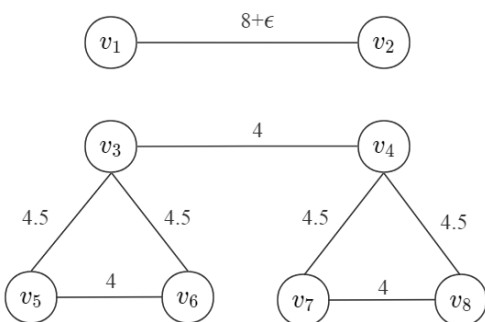

Figure 8: The graph is allocated to two agents with identical valuations. The allocation maximizing Nash social
welfare fails to guarantee EF1.

Although Proposition D.1 is disappointing in general, an allocation that maximizes Nash social
welfare is PO and EF2, and when the valuations are identical, it ensures 2/3-EF1.

**Proposition D.2.** *A Nash social welfare maximizing allocation is PO and EF2 for any matching-*
*induced valuations, and is 2/3-EF1 if the valuations are identical.*

*Proof.* We only prove for the case of two agents, and the proof for arbitrary number of agents is
the same. Given any graph $G = (V, E)$, suppose $(V_1, V_2)$ is an allocation that maximizes the Nash
social welfare. Denote by $G_1 = (V_1, E_1)$ and $G_2 = (V_2, E_2)$ the induced subgraphs by $V_1$ and $V_2$
respectively. Denote by $M_1$ and $M_2$ the maximum matchings in $G_1$ and $G_2$. To show $(V_1, V_2)$ is
EF2, we can regard the edges in $M_1$ and $M_2$ as items and by the result from [Caragiannis *et al.*,

2019], the allocation must be envy-free up to one edge, which guarantees EF2, since deleting one edge can be achieved by deleting two nodes.

Next we show the allocation is 2/3-EF1 for identical valuations. For the sake of contradiction, suppose that for any $v \in V_2$,

$$u(V_1) = w(M_1) < \frac{2}{3} \cdot u(V_2 \setminus \{v\}) \leq \frac{2}{3} \cdot w(M_2).$$

By Lemma A.4, $|M_2| \geq 3$ and thus the smallest edge in $M_2$ denoted by $e$ has weight at most $1/3 \cdot w(M_2)$. Thus

$$w(M_1) + w(e) < \frac{2}{3} \cdot w(M_2) + \frac{2}{3} \cdot w(M_2) = w(M_2). \tag{9}$$

Consider a new allocation by assigning edge $e$ to agent 1. The resulting maximum matchings in each subgraph becomes $M_1 \cup \{e\}$ and $M_2 \setminus \{e\}$. The Nash social welfare of the new allocation is

$$
\begin{aligned}
& w(M_1 \cup \{e\}) \cdot w(M_2 \setminus \{e\}) \\
&= (w(M_1) + w(e)) \cdot (w(M_2) - w(e)) \\
&= w(M_1) \cdot w(M_2) + (w(M_2) - w(M_1) - w(e)) \cdot w(e) \\
&> w(M_1) \cdot w(M_2),
\end{aligned}
$$

where the inequality is by (9). However, this is a contradiction with the fact that $(V_1, V_2)$ maximizes the Nash social welfare. $\square$

# E  An Improved MMS Allocation Algorithm for Two Homogeneous Agents

When there are only two agents, Algorithm 1 can be refined as shown in Algorithm 6, and the approximation ratio can be improved to 2/3. Algorithm 6 is similar with Algorithm 1; we first compute a maximum matching $M^*$ and a max-min partition $(M_1, M_2)$ with $w(M_1) \geq w(M_2)$. If $|M_1| \geq 2$, we output the corresponding allocation. Otherwise, in graph $G$, we directly delete the edge that $M_1$ contains. We repeat the above procedure until all edges are removed.

---

**Algorithm 6:** Max-Min Allocation for 2 Agents

---

**Input:** Instance $\mathcal{I} = (G, N, u)$ with $G = (V, E; w)$.
**Output:** Allocation $\mathbf{X} = (X_1, X_2)$.
 1: Find a maximum matching $M^*$ in $G$. Denote by $V'$ the set of unmatched vertices by $M^*$.
 2: Find the greedy partition $(M_1, M_2)$ of edges in $M^*$ such that $w(M_1) \geq w(M_2)$.
 3: Let $Max = w(M_2)$.
 4: Set $X_1 = V(M_1)$.
 5: Set $X_2 = V(M_2) \cup V'$.
 6: **while** $w(M_1) > 2w(M_2)$ **do**
 7:     By Lemma 3.3. $M_1$ must contain only one edge. Suppose $M_1 = \{e^*\}$.
 8:     Delete edge $e^*$.
 9:     Re-compute a maximum matching $M^*$.
10:     Re-set $V'$ to be unmatched vertices by $M^*$.
11:     Re-compute the greedy partition $(M_1, M_2)$ of $M^*$ such that $w(M_1) \geq w(M_2)$.
12:     **if** $Max < w(M_2)$ **then**
13:         $Max = w(M_2)$.
14:         Set $X_1 = V(M_1)$.
15:         Set $X_2 = V(M_2) \cup V'$.
16:     **end if**
17: **end while**
18: Output allocation $(X_1, X_2)$.

---

**Theorem E.1.** *Algorithm 6 outputs an allocation that is* $2/3$*-approximate max-min fair in polynomial time* .

*Proof.* Given an Instance $\mathcal{I} = (G, N, u)$ with $G = (V, E; w)$. Denote by $O = (O_1, O_2)$ the optimal solution, where $u(O_1) \geq u(O_2)$ and $\text{opt}(\mathcal{I}) = u(O_2)$. The first time when we reach the **while** loop, if $w(M_1) \leq 2 \cdot w(M_2)$, allocation $(X_1, X_2) = (V(M_1), V(M_2) \cup V')$ has been output. By Algorithm 6, we have

$$w(M_1) \geq u(O_1) \geq u(O_2) \geq w(M_2).$$

Moreover,

$$
\begin{aligned}
w(M_2) &\geq \frac{1}{3} \cdot (w(M_1) + w(M_2)) \\
&\geq \frac{1}{3} \cdot (u(O_1) + u(O_2)) \\
&\geq \frac{1}{3} \cdot 2 \cdot u(O_2) = \frac{2}{3} \cdot u(O_2).
\end{aligned}
$$

We move into the **while** loop if $w(M_1) > 2 \cdot w(M_2)$. In such case, $M_1$ contains only one edge, i.e., $|M_1| = 1$. Suppose $M_1 = \{e^*\}$. There are two subcases:

- Case 1: $e^* \in O_1 \cup O_2$

- Case 2: $e^* \notin O_1 \cup O_2$

First. consider $|M_1| = 1$ and $e^* \in O_1 \cup O_2$. We have: $e^* \in O_1$ and $w(M_2) = u(O_2) = \text{opt}(\mathcal{I})$. The optimal solution has been found and recorded. Therefore, the approximation ratio of the max-min partition is 1. Then the **while** loop is executed for the next round. When $e^* \notin O_1 \cup O_2$, edge $e^*$ is deleted. Let $(M_1', M_2')$ be the greedy partition after deleting edge $e^*$. Then there are two subcases:

- Subcase 1: $|M_1'| \geq 2$

- Subcase 2: $|M_1'| = 1$

For Subcase 1, we will get out of the **while** loop and a $2/3$-approximate max-min allocation has been determined. For Subcase 2, the **while** loop is executed for the next round. Since we are not sure whether $e^* \in O_1 \cup O_2$, the **while** loop is executed for at most $O(m^2)$ rounds ($m$ is the number of nodes in graph $G(V, E)$). The output $(X_1, X_2)$ is at least $2/3$-approximate max-min fair allocation. Thus, the theorem holds. $\qquad\square$

**Lemma E.2.** *Algorithm 6 outputs an allocation that is $1/2$-approximate max-min fair by eliminating at most two edges.*

*Proof.* Given an Instance $\mathcal{I} = (G, N, u)$ with $G = (V, E; w)$. Denote by $O = (O_1, O_2)$ the optimal solution before eliminating any edge, where $u(O_1) \geq u(O_2)$ and $\text{opt}(\mathcal{I}) = u(O_2)$. Initially, under the maximum matching $M'$, we find the greedy max-min partition $(M_1, M_2)$ such that $w(M_1) \geq w(M_2)$. If $|M_1| \geq 2$, by Lemma 3.3, $(M_1, M_2)$ is a $1/2$-approximation max-min partition. Next, consider that $M_1$ contains only one edge. Suppose $M_1 = \{e_1\}$ and $e_1 \notin O_1 \cup O_2$ (otherwise, by Case 1 in Theorem E.1, $w(M_2) = u(O_2) = \text{opt}(\mathcal{I})$. The optimal solution has been found). If we eliminate edge $e_1$, under the re-computed maximum matching, we find the greedy max-min partition $(M_1', M_2')$. Let $O' = (O_1', O_2')$ denote the optimal solution after eliminating edge $e_1$, where $u(O_1') \geq u(O_2')$ and $\text{opt}'(\mathcal{I}') = u(O_2')$. $M_1' = \{e_1'\}$ and $e_1' \notin O_1' \cup O_2'$. We first show that the two edges $e_1$ and $e_1'$ have one common endpoint. By Algorithm 6,

$$w(e_1) > u(O_1) \geq u(O_2) > w(M_2),$$

and

$$w(e_1') > u(O_1') \geq u(O_2') > w(M_2').$$

Hence, $w(e_1') > w(M_2)$. Furthermore,

$$w(e_1) + w(e_1') > w(e_1) + w(M_2).$$

Therefore, edges $e_1$ and $e_1'$ make a maximum matching, which implies that there exists an allocation to improve the max-min value from $u(O_2)$ to $w(e_1')$. It results in a contradiction. Hence, the two edges $e_1$ and $e_1'$ have one common endpoint. Furthermore, $e_1 \notin O_1 \cup O_2$ (otherwise, suppose

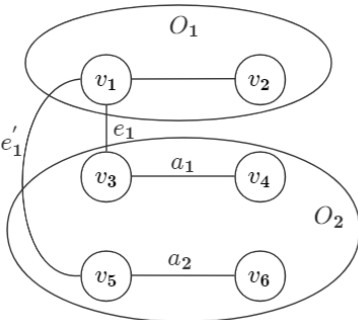

Figure 9: The graph is allocated to two agents with identical valuations. Two large edges $e_1$ and $e_1'$ have one common endpoint and their other endpoints connect two distinct edges.

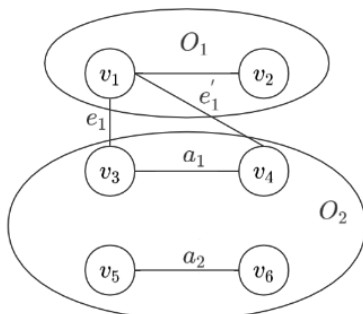

Figure 10: The graph is allocated to two agents with identical valuations. Two large edges $e_1$ and $e_1'$ have one common endpoint and their other endpoints are the two endpoints of another same edge.

$e_1 \in O_1$. Then $O_1$ can be replaced by edge $e_1$ to get larger welfare, which makes a contradiction). Therefore, there are two cases. First, we consider Case 1 (as shown in Figure 9): suppose $a_1$ and $a_2$ are two edges in $O_2$, and $w(a_1) \geq w(a_2)$. Denote $O_2^* = O_2 / \{a_1, a_2\}$. Thus

$$\max(w(a_1), w(a_2)) \geq \frac{1}{2}(w(a_1) + w(a_2)),$$

and then

$$w(a_1) + u(O_2^*) \geq \frac{1}{2} u(O_2).$$

Accordingly, either

$$w(M_2) \geq w(a_1) + u(O_2^*)$$

or

$$w(M_2') \geq w(a_1) + u(O_2^*)$$

holds; otherwise, replacing $M_2$ or $M_2'$ with $a_1 \cup O_2^*$ can make a matching with larger welfare. Therefore, $w(M_2) \geq 1/2 \cdot u(O_2)$ or $W(M_2') \geq 1/2 \cdot u(O_2)$. Next, we consider Case 2 (as shown in Figure 10). Suppose after the two edges $e_1$ and $e_1'$ have been deleted, under the new maximum matching $M''$, we find the greedy partition $(M_1'', M_2'')$. If $|M_1''| \geq 2$, by Theorem E.1, the 2/3-approximate max-min partition can be found. If $|M_1''| = 1$, then

$$w(M_2'') \geq w(a_1) + O(B') \geq \frac{1}{2} u(B_2).$$

Thus a 1/2-approximate max-min partition has been found, and the lemma holds. $\qquad\square$

# F    Examples

## F.1    An Example where Envy-cycle Elimination Algorithm does not Work

Consider a path of four nodes $v_1 \to v_2 \to v_3 \to v_4$, and two agents have the same weight 1 on all three edges $(v_1, v_2)$, $v_2, v_3$ and $v_3, v_4$. By ency-cycle elimination algorithm, we may first allocate the items in the following order: $v_1$ to agent 1, then $v_2$ to agent 2, then $v_3$ to agent 1 and finally $v_4$ to agent 2. Note that $u_1(\{v_1, v_3\}) = v_2(\{v_2, v_4\}) = 0$, however, the optimal social welfare is 2 by allocating $\{v_1, v_2\}$ to agent 1 and $\{v_3, v_4\}$ to agent 2. Thus the approximation ratio of the social welfare is unbounded.

## F.2    $e_1$ May not Have the Largest Weight

In the execution of Algorithm 1, the edge $e_1$, which is the only edge in $M_1$, may not have the largest weight. Consider an instance with two agents and the graph is shown in Figure 11. By Algorithm 1, in the first round of the *while* loop, we find a maximum matching, say, $M^* = \{e_{12}, e_{34}, e_{56}, e_{78}\}$. The greedy partition of $M^*$ is $M_1 = \{e_{34}\}$, $M_2 = \{e_{56}\}$, $M_3 = \{e_{12}, e_{78}\}$. Since $w(M_3) = 4 < 1/2 \cdot w(M_1) = 32$, $H = \{e_{34}, e_{45}, e_{56}\}$, which contains all the edges with at least $w(e_{34}) = 32$. Noe that in this case, $e_1$ does not have the highest 64.

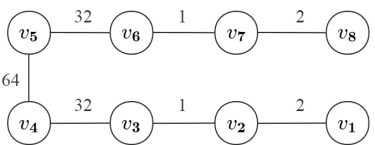

Figure 11: The graph is allocated to three agents with identical valuations.