# OpenReview forum: "Graphical Resource Allocation with Matching-Induced Utilities"
_NeurIPS.cc/2022/Conference — NeurIPS 2022 Submitted_

### Official Review · Reviewer_JkrW · 2022-07-10

**Rating:** 8
**Confidence:** 4
**Soundness:** 3 good
**Presentation:** 4 excellent
**Contribution:** 4 excellent

**Summary:**

This paper studies mechanisms for the graphical resource allocation problem with two types of fairness: approximate maximin share (MMS) and envy-freeness up to one item (EF1). The problem is defined as follows: the resources are represented as nodes in an undirected graph, and the goal is to allocate (or partially allocate) the nodes to n agents. Each agent has an edge weight function for each edge in the graph, and an agents’ utility on one partition is the weight of a maximum matching in the subgraph induced by the nodes in the partition.

The authors first consider a setting of homogeneous agents that all agents have identical valuation functions. They show that a naive greedy partition is an MMS allocation for unweighted graphs, but does not have any bounded approximation guarantee in more general cases. Then they propose an algorithm that computes a ⅛-MMS allocation in polynomial time. Because any bounded approximation ratio for MMS fairness could have unbounded social welfare loss, EF1 is also studied to preserve high social welfare while having fairness guarantee. Theorem 3.6 shows that a polynomial time algorithm returns an EF1 allocation with an approximate optimal social welfare.

For agents that have heterogeneous valuation functions, the authors first show some negative results that no algorithm has bounded approximation for MMS and no algorithm has better than 1/n approximation of social welfare for EF1 fairness. Then they consider two special cases: 1, when the agents have binary weight functions, Algorithm 3 returns an EF1 allocation with 1/3 -approximate social welfare. 2, In the case that there are only two agents, Algorithm 4 returns an EF1 allocation with a tight ⅓ approximation ratio to the optimal social welfare.


**Questions:**

* Are there more closely related works that combine graph partition and fair division together? See my comments in “Strengths And Weaknesses”.
* Could you give some lower bounds of the results? See my comments in “Limitations”.
* Page 5, line 198: “The approximation ratio is at least 1/4”. By Lemma 3.3, why isn’t the approximation ratio ½?


**Limitations:**

One thing could make this paper stronger is to have more lower bound results. Theorem 4.4 showed that the 1/3 -approximation to social welfare with EF1 is tight, but most of the results in this paper are without lower bounds. It would be interesting to see some lower bounds, even loose ones. The authors have addressed this in Section 5 “Future Directions”.

**Strengths And Weaknesses:**

* Originality: This work proposes a setting of fair division on indivisible graphical items where agents have combinatorial valuations. Two commonly used fairness concepts MMS and EF1 are studied. Related works on graph partition and fair division are properly cited. I do wonder if there are more closely related works that combine graph partition and fair division together?
* Quality: The notations are well defined. The theorems / lemmas are neatly presented and proved.
* Clarity: This paper is well written and organized. Some minor comments:
  * Page 1, line 15: “ensuring constant fraction” -> ensuring a constant fraction
  * Page 2, line 49: “significant amount of” -> a significant amount of
  * Page 2, line 53: “widely accepted and studies” -> widely accepted and studied
  * Page, line 83: “for two-agent case” -> for the two-agent case
  * Page 9, line 295 “smallest ” -> the smallest; page 9 line 302: “second smallest ” -> the second smallest
* Significance: The results in both the homogeneous and heterogeneous settings are interesting and could be building blocks for future works in this line.

---

> ### Author Response · Authors · 2022-08-02
> **Response to Reviewer JkrW**
>
> We appreciate the reviewer's constructive comments.
> We have followed all the comments to polish the paper, and fixed the typos.
> In the following, we answer the reviewer's questions.
>
> Question: Related works on graph partition and fair division are properly cited. I do wonder if there are more closely related works that combine graph partition and fair division together?
>
> Answer: We thank the reviewer for this question. Actually, there is a rich line of research papers that combine graph partition and fair division; see, e.g., [Bouveret et al., 2017; Suksompong, 2019; Bilò et al., 2019; Igarashi and Peters, 2019].
> However, in these works where the items are the vertices, the graph is used to restrict that the allocation to each agent forms a connected subgraph in the given graph. However, the value for the subgraph is still additive, i.e., summing up the values for the vertices/items.
> As far as we know, the case when the value of an allocation depends on its graphical structures (like matching in our paper) has not been studied yet.
>
> Question: Could you give some lower bounds of the results?
>
> Answer: This is a very good question!
>
> For the case of homogeneous agents, despite significant effort, we were not able to provide any lower bounds.
> Actually, we conjecture that there always exists an EF1 allocation that achieves the optimal social welfare.
>
> For the case of heterogeneous agents,  below we provide an instance showing a lower bound of $2$ when the agents have binary weight functions.
> Consider a square shown in Figure 3 in Appendix B.1. The maximum social welfare can be achieved by allocating the whole graph to one of the agents, resulting in a social welfare of 2. However, this allocation is not EF1.
> Actually, to guarantee EF1 in this instance, we can allocate at most three vertices to one agent, resulting in a social welfare of 1.
>
> This lower bound instance is also included in the revised submission.
>
> Question: Page 5, line 198, “The approximation ratio is at least 1/4”. By Lemma 3.3, why isn’t the approximation ratio 1/2?
>
> Answer: We apologize for being unclear here.
> The reviewer is correct that by Lemma 3.3, we lose an approximation of 1/2 when we use the greedy partition.
> However, before finding the maximum matching and its greedy partition, we first rounded down all the edge values to the closest power of 2, which makes the approximation ratio lose another factor of 2 by Lemma 3.4, and thus the approximation ratio is 1/4.
> We refined this sentence in the revised submission.

---

> > ### Comment · Reviewer_JkrW · 2022-08-08
> > **Thank you for the response**
> >
> > Thank you for the response and revision of the paper!

---

### Official Review · Reviewer_N528 · 2022-07-13

**Rating:** 4
**Confidence:** 4
**Soundness:** 3 good
**Presentation:** 1 poor
**Contribution:** 2 fair

**Summary:**

This paper deals with a novel variant of the fair division problem which deals with the allocation of indivisible items to multiple agents with complementarities. Each agent’s valuations are given by a weighted undirected graph where there is a node for each item and the edge weights correspond to the value the agent receives when the items corresponding to the end points of the edge are paired together. The value for a bundle of items is given by the total value of the maximum weight matching that can be obtained in the subgraph induced by the nodes corresponding to the items in the bundle. The authors motivate this setting by the example of departments (agents) in an engineering firm having to pair up the engineers (resources / items) assigned to the department, and other settings where employees must be paired to achieve tasks that generate positive value.

The main results deal with computing fair, i.e., envy-free up to one item [EF1] and maximin-share [MMS], and efficient, i.e., social welfare maximizing, allocations of items to agents. The authors provide several interesting positive and negative results on efficiently computing allocations that guarantee a (fraction of) agents’ MMS share or total social welfare under constraints such as guaranteeing EF1 under settings with homogeneous (all agents have identical valuation functions / graphs) or heterogeneous (agents have possibly different valuations) agents.

**Questions:**

No significant questions. Please see detailed comments.

**Limitations:**

The authors do not identify the societal impact of this work. [Neutral]

**Strengths And Weaknesses:**

Strengths:

[Conceptual contributions]
+ The setting is novel and possibly interesting to the overall research community studying fair division of indivisible items.
+ The proposed model presents interesting directions for future work, as the authors note. In particular, other valuation functions that depend on the subgraph induced by the items in a bundle open several interesting possibilities.

[Technical contributions]
+ The theoretical results are non-trivial and the proof sketches in the main body are convincing.
+ The results in the paper are therefore a good first step.

Weaknesses:
- [Relevance] I am struggling to find the relevance to NeurIPS, although I will leave this matter for later discussion, and have not allowed it to affect my final recommendation at this time.
- [Motivation] The motivating examples are unconvincing and lack any citations or references. For example, the authors claim that the motivating problem of a company with departments who have preferences over how to pair engineers is a real-world scenario. I think I understand what they refer to, but the authors provide no formal references or citations.
- [Significance] While the technical results are interesting, due to the poor presentation, the significance of the results is not clear.

[Presentation]
- Poor quality of writing overall. See detailed comments below.
- Several theorem statements are imprecise. Detailed comments below.
- Several arguments are vague. I understand that there is a space limitation, but perhaps this can be revised in a future iteration. Detailed comments below.
- There are some interesting technical results, but perhaps the most interesting and significant results can be highlighted more prominently. In particular, the results for heterogeneous agents on the "price of EF1" in terms of social welfare and impossibility of guaranteeing any constant fraction of the MMS share are interesting negative results.
- A conjecture is presented in the Section 1.1 that an EF1 and social welfare maximizing allocation always exists but no argument is provided in the relevant Section 3.2 [I apologize if I missed it, and am happy to be corrected].

Detailed comments:

[Writing]
- L1: Motivated by the real ... - > Motivated by real ...
- L4: We care both ... -> Consider revising
- L5: Regarding MMS ... -> Consider revising
- L5: ... does not admit finite ... -> ... does not admit a finite ...
- L20-22: Needs more explanation or context of what is meant by a graphic resource (especially i did not understand how graph neural networks are relevant here)
- L23-26: Please cite some references (e.g. related work) for this motivating example. Also, the work on hedonic games may be relevant to discuss at some point.
- L52-56: EF1 was proposed by Budish, 2011 (later) and proven to always exist by Lipton et al., 2004. This is confusing. I am familiar with the literature. Perhaps there is a better way to write this?
- L127: ... if the weight function is highlighted. Consider revising this sentence.
- L128-132: The imaginary experiment leaves some questions, and can perhaps be made more precise. From the description, it is not clear if the agent has knowledge of other agents' valuations and therefore of how other agents will pick the partitions. AFAIK, MMS is defined for the worst-case. Perhaps it is better to describe it as the maximum value the agent can have for the worst partition in any n-partitioning. And not involve the selections of other agents.
- L136: ... if for all agent ... -> ... if for all agents ...
- L137: ... is called MMS fairness ... -> ... is called MMS fair ...
- L143: Accordingly, people mostly ... - > Please consider revising
- L145: The line claims that EF1 is widely accepted and studied. But no citations or references are provided. I suggest that this be motivated and discussed earlier (e.g. Introduction or Related Works) and not repeated here, and only formal definitions and helpful examples be provided in the Preliminaries.
- L199: ... decent ... -> Not clear what this means
- L200: ... far smaller ... -> Consider revising
- Algorithm 3: The comments for the different cases are not made consistently. Please consider revising.
- L299: ... if so we do exchange ... -> Please consider revising.
- L302: ... and so on so forth. -> ... and so on.

[Theorem statements]
- L164: I believe this only applies to the case of homogeneous agents, which is not clear from the theorem statement.
- L222: ... two cases happens. -> ... two cases hold true. (or something similar)
- L268 and L270: Please consider specifying that this is for the case of heterogeneous agents.
- L297: ... j envies i ... -> ... j envy i ...
- L304: Please clarify whether this result only holds for binary weights.

[Arguments]
- L158-160: Please consider making this more precise and providing a brief argument and cite the (variant of the) Partition problem.
- L169: This task is NP-hard ... -> Why? [I think I know why] Please specify and consider rewriting to avoid repetitions.
- L188-190: I don't think this statement is as helpful as it was perhaps intended to be. It is too vague.
- L211: Not clear what optimal means here (could be MMS or max SW). Is O any MMS allocation?
- L226-227: Please consider revising this. It is hard to parse this sentence.
- L245: In the equation, please consider revising: for any v ... -> for every v ... Currently, it reads like EFX
- L254-261: Please consider using a proof sketch environment if possible and avoid mixing descriptions of the algorithm with the proof.
- L271 and Theorem 4.1: Please consider including the here, since it is indeed a strong negative result.
- L293: It is not clear why this matters. Please consider revising and illustrating this with an example.

[Others]
- Algorithm 1: It appears that e_1 is always the highest weight edge (due to it being the only edge in M_1) and in line 5, H is the set of all highest weight edges.
- Line 290: Please consider providing an example here.

---

> ### Author Response · Authors · 2022-08-02
> **Response to Reviewer N528**
>
>
> We appreciate the reviewer's constructive comments.
> Before we answer the reviewer's questions, we want to say that NeurIPS is a suitable venue for our paper because it does fall under the umbrella of the track “Theory (e.g., control theory, learning theory, algorithmic game theory)” in Call for Papers. We can see that NeurIPS is becoming more open to Algorithmic Game Theory, Combinatorial Optimization works and other interdisciplinary research, and fair division is a typical problem therein.
> This can also be verified by the following papers that were accepted in the past few years.
>
>   Fair Scheduling for Time-dependent Resources (NeurIPS 2021)
>
>   Explainable Voting (NeurIPS 2020)
>
>   Exploring Algorithmic Fairness in Robust Graph Covering Problems (NeurIPS 2019)
>
>   Balancing Efficiency and Fairness in On-Demand Ridesourcing (NeurIPS 2019)
>
>  A Graph-Theoretic Additive Approximation of Optimal Transport (NeurIPS 2019)
>
> Next we provide a point-by-point response to the reviewer's comments.
>
> [writing]
>
>  L1: Motivated by the real $\ldots$ $\rightarrow$ Motivated by real $\ldots$
>
>  L4: We care both $\ldots \rightarrow$ We care about both
>
>  L5: Regarding MMS $\ldots$ $\rightarrow$ Regarding MMS fairness
>
>  L5: $\ldots$ does not admit finite $\ldots \rightarrow$ $\ldots$ does not admit a finite $\ldots$
>
>  L20-22: Needs more explanation or context of what is meant by a graphic resource (especially i did not understand how graph neural networks are relevant here)
>
> These papers are not purely on fair resource allocation but all of them consider graphic structures.
>
>    For example, [Ying et al., 2019] formulated a model that considers the mutual information between a Graph Neural Network's prediction and the distribution of possible subgraph structures in a given graph.
>     Our intention was to use these papers to motivate why we are interested in graph structures.
>     Since they are not about fair resource allocation, we did not include more details.
>
> We have rewritten this part in the revised version.
>
>
> L23-26: Please cite some references (e.g. related work) for this motivating example. Also, the work on hedonic games may be relevant to discuss at some point.
>
>  We wrote the motivating example and cite the corresponding papers in the revised version as follows.
>
>  Peer Instruction (PI) has been shown to be an effective learning approach based on a project conducted at Harvard University, and one of the simplest ways to implement PI is to pair the students [Crouch C H et al. 2001].
>         Consider the situation when we partition students to advisors, where the advisors will adopt PI for their assigned students. Note that the advisors may hold different perspectives on how to pair the students based on their own experience and expertise, and they want to maximize the efficiency of conducting PI in their own assigned students. How should we assign the students fairly to the advisors? How can we maximize the social welfare among all (approximately) fair assignments? Our results shed light on solving these two questions.
>
> [Crouch C H et al. 2001] Crouch C H , Mazur E. Peer Instruction: Ten years of experience and results[J]. American Journal of Physics, 2001, 69(9):970-977.
>
>  We also added references on hedonic games in the second paragraph of introduction.
>
> L52-56: EF1 was proposed by Budish, 2011 (later) and proven to always exist by Lipton et al., 2004. This is confusing. I am familiar with the literature. Perhaps there is a better way to write this?
>
> We rewrite this as follows:
>
>  We added one footnote here: The algorithm in [Lipton et al., 2004] was originally published in 2004 with a different targeting property. In 2011, Budish [2011] formally proposed the notion of EF1 fairness.
>
>
> L127: ... if the weight function is highlighted. Consider revising this sentence.
>
> We rewrote the sentence as follows: A problem instance is denoted by $\cI= (G, N)$. When we want to highlight the weight function, $w$ is also included as a parameter, i.e., $\cI = (G, N, w)$.
>
> L128-132: The imaginary experiment leaves some questions, and can perhaps be made more precise. From the description, it is not clear if the agent has knowledge of other agents' valuations and therefore of how other agents will pick the partitions. AFAIK, MMS is defined for the worst-case. Perhaps it is better to describe it as the maximum value the agent can have for the worst partition in any n-partitioning. And not involve the selections of other agents.
>
> We rewrote this intuition in the counterpart of introduction. We removed this intuition here and directly introduced the formal definition.
>
> L136: $\ldots$ if for all agent $\ldots \rightarrow \ldots$ if for all agents $\ldots$
> L137: $\ldots$ is called MMS fairness $\ldots \rightarrow \ldots$ is called MMS fair $\ldots$

---

> > ### Author Response · Authors · 2022-08-02
> > **Response to Reviewer N528**
> >
> > L143: Accordingly, people mostly $\ldots$ $\rightarrow$ Accordingly, researchers have sought $\ldots$
> >
> > L145: The line claims that EF1 is widely accepted and studied. But no citations or references are provided. I suggest that this be motivated and discussed earlier (e.g. Introduction or Related Works) and not repeated here, and only formal definitions and helpful examples be provided in the Preliminaries.  $\rightarrow$ Done.
> >
> > L199: $\ldots$ decent $\ldots \rightarrow$ constant
> >
> > L200:$\ldots$ far smaller $\ldots \rightarrow$  the utility of smallest bundle is much less than the largest bundle.
> >
> >  Algorithm 3: The comments for the different cases are not made consistently. Please consider revising.
> >
> > We renamed the third case by  {Skip the Current Agent}
> >
> > L299: $\ldots$if so we do exchange $\rightarrow$ if so we execute exchange procedure.
> >
> > L302: $\ldots$ and so on so forth. $\rightarrow$ Done.
> >
> > [Theorem statements]
> >
> > L164: I believe this only applies to the case of homogeneous agents, which is not clear from the theorem statement. $\rightarrow$ Done.
> >
> > L222: $\ldots$ two cases happens. $\rightarrow$ Done.
> >
> > L268 and L270: Please consider specifying that this is for the case of heterogeneous agents. $\rightarrow$ Done.
> >
> > L297: $ \ldots$ j envies i $\ldots \rightarrow$ Done.
> >
> > L304: Please clarify whether this result only holds for binary weights.   The title of Algorithm 3 shows that this result only holds for binary weights. $\rightarrow$ It holds for binary weights and we clarified this in the revised submission.
> >
> > [Arguments]
> >
> > L158-160: Please consider making this more precise and providing a brief argument and cite the (variant of the) Partition problem.
> >
> > $\rightarrow$ Done.
> >
> > L169: This task is NP-hard $\ldots \rightarrow$ Why? [I think I know why] Please specify and consider rewriting to avoid repetitions.
> >
> >  $\rightarrow$  Thanks for pointing this out, and we will explain and add citation here.
> >
> >  A special case of this problem is a set of independent edges to be partitioned into $n$ bundles so that the minimum bundle has largest weight. This is essentially a partition problem, which is NP-hard.
> >
> > L188-190: I don't think this statement is as helpful as it was perhaps intended to be. It is too vague.
> >
> > $\rightarrow$ Done!
> >
> > L211: Not clear what optimal means here (could be MMS or max SW). Is O any MMS allocation?
> >
> > $\rightarrow$  We apologize for being unclear here.
> >  The optimality is for MMS, and thus $O$ is one allocation that maximizes the value of the smallest bundle.
> >
> > L226-227: Please consider revising this. It is hard to parse this sentence.
> >
> > $\rightarrow$  By Claim 3.5, the {\bf while} loop will not execute Case 2 or it executes Case 1 for several times and then Case 2 for exactly once.
> >
> > L245: In the equation, please consider revising: for any v $\ldots \rightarrow$ for every v $\ldots$ Currently, it reads like EFX
> >
> > $\rightarrow$ Done.
> >
> >
> >
> > L271 and Theorem 4.1: Please consider including the here, since it is indeed a strong negative result.
> >
> > $\rightarrow$ Done.
> >
> > L293: It is not clear why this matters. Please consider revising and illustrating this with an example.
> >
> > $\rightarrow$ In the revised version, we included the following example, where the ency-cycle elimination algorithm does not have bounded approximation ratio.
> >
> > Consider a path of four nodes $v_1 \to v_2 \to v_3 \to v_4$, and two agents have the same weight 1 on all three edges $(v_1,v_2)$, $v_2,v_3$ and $v_3,v_4$.
> > By ency-cycle elimination algorithm, we may first allocate the items in the following order: $v_1$ to agent 1, then $v_2$ to agent 2, then $v_3$ to agent 1 and finally $v_4$ to agent 2. Note that $u_1(\{v_1,v_3\}) = v_2(\{v_2, v_4\}) = 0$, however, the optimal social welfare is 2 by allocating $\{v_1,v_2\}$ to agent 1 and $\{v_3,v_4\}$ to agent 2. Thus the approximation ratio of the social welfare is unbounded.
> >
> > [Others]
> >
> > Algorithm 1: It appears that $e_1$ is always the highest weight edge (due to it being the only edge in $M_1$) and in line 5, H is the set of all highest weight edges.
> >
> > $\rightarrow$ Actually, $e_1$ may not have highest weight. We provided one example in the appendix of the revised submission.
> >     The high-level idea is that the edge with highest weight may not appear in a maximum matching.
> >     But it does not matter whether $e_1$ has highest weight or not since all the edges whose weights are no smaller than $e_1$'s will be degraded to a half of  $e_1$'s weight.
> >
> > Line 290: Please consider providing an example here.
> >
> > $\rightarrow$ We included the example in the revised submission.

---

### Official Review · Reviewer_Vymq · 2022-07-16

**Rating:** 3
**Confidence:** 5
**Soundness:** 3 good
**Presentation:** 3 good
**Contribution:** 1 poor

**Summary:**

The paper considers a new variant of fair resource allocation problem: given a set of resources as vertices of a graph, the goal is to partition the resources among N agents -- an agent utility is given by the maximum matching on the induced subgraph of his partition. They consider two fairness measures on the utilities received by each agent : max min share (MMS) - where the minimum utility of agents is maximized, envy free upto one item (EF1) - where the no agent envies others' allocation after removing upto one item.

The main results of the work are the following: a 1/8 approximation algorithm for MMS version of the problem where all the agents are identical and a constant approximation for the EF1 variant. The work also gives some hardness results and better approximation in certain special cases.

**Questions:**

As mentioned above, the authors must address two issues with the paper: are there any real motivating examples/usecases for the problem? Secondly, what is the key central novel contribution of the work.

**Strengths And Weaknesses:**

The paper is reasonable well written and easy to follow. Unfortunately, there are multiple severe issues with it.

Firstly, I am really not convinced of the motivation behind defining the problem. The motivating example of "pairing up employees" is really weak and sounds made-up to fit the problem definition.

In spirit of a theoretical work, the motivation can be ignored if there are sufficiently interesting results. Unfortunately, the algorithms and analysis is just a rehashing of existing ideas.

For instance, for the MMS version of the problem, the approach is to find a maximum matching followed by a decomposition of the independent edges. This is really the first approach one can think of and in fact, putting aside the paper, I came up with it myself in 5 mins. The paper then elaborates a way to decompose the independent edges in N parts. But this is a standard problem in approximation algorithms used for example load-balancing and several other scheduling problems. Indeed, in bin packing, adding items to lowest occupied bin yields a simple 2-approximation and these results are known from the 1960s. Spending 2-3 pages on this result really underlines the technical inferiority of the work.

---

> ### Author Response · Authors · 2022-08-02
> **Response to Reviewer Vymq**
>
> Question: Are there any real motivating examples/usecases for the problem?
>
> Answer: We can consider the following real-world example to motivate our model, and we have included this example in the revised version.
>
> Peer Instruction (PI) has been shown to be an effective learning approach based on a project conducted at Harvard University, and one of the simplest ways to implement PI is to pair the students [Crouch C H et al. 2001].
> Consider the situation when we partition students to advisors, where the advisors will adopt PI for their assigned students.
> Note that the advisors may hold different perspectives on how to pair the students based on their own experience and expertise, and they want to maximize the efficiency of conducting PI in their own assigned students.
> How should we assign the students fairly to the advisors?
> How can we maximize the social welfare among all (approximately) fair assignments?
> Our results shed light on solving these two questions.
>
> [Crouch C H et al. 2001] Crouch C H , Mazur E. Peer Instruction: Ten years of experience and results[J]. American Journal of Physics, 2001, 69(9):970-977.
>
> Question: What is the key central novel contribution of the work?
>
> Answer: EF and MMS related fairness are being very widely investigated but most of the research is focusing on the additive valuations. Although there are several exceptions where submodular (or XoS and subadditive) valuations are studied, general combinatorial especially graphic structural functions have not been studied so far. Our key  novel contribution is to initiate this line of research and uncover some interesting future directions.
>
> Comment: "{ ..., the approach is to find a maximum matching followed by a decomposition of the independent edges. This is really the first approach one can think of and in fact, putting aside the paper, I came up with it myself in 5 mins.}"
>
> Answer: We agree with the reviewer that the first attempt to solve this problem is to find a maximum matching followed by a decomposition of the independent edges.
> The reviewer is smart and came up with this general idea quickly; however, as we highlighted in Figure 1, this idea can perform arbitrarily bad.
>
> As we have shown in the paper, the difficulty of our algorithm is how to gradually modify the edge weights until it produces a good matching (which may not be the maximum) and can be partitioned so that the minimum bundle is large.
>
> Comment: "{..., Spending 2-3 pages on this result really underlines the technical inferiority of the work.}"
>
> Answer: First, we only used a single short paragraph to explain why the maximum matching + decomposition idea does not work, and the 2-3 pages are for introducing how to revise the edge weights and find a proper (decomposable) matching that may not be a maximum one, and proving its approximation ratio.
>
> Second, as we mentioned the techniques here are not straightforward, and we believe it is necessary to rigorously prove all statements to ensure the correctness.
>
> Moreover, we want our paper to be friendly to the audience of NeurIPS who are not theoreticians.
>
> Finally, we want to use the homogeneous setting to warm up the readers in order to get a good understanding of our model and the techniques behind.

---

### Official Review · Reviewer_UCR6 · 2022-07-19

**Rating:** 5
**Confidence:** 3
**Soundness:** 4 excellent
**Presentation:** 4 excellent
**Contribution:** 2 fair

**Summary:**

This paper studies a variant of the indivisible item allocation problem. In this variant, the items are vertices of an edge-weighted graph. An agent’s valuation on a bundle is given by the weight of the maximum weight matching on the subgraph induced by the vertices in the bundle. In the homogeneous setting, the same edge-weighted graph is defined as the valuation of all the agents. In the heterogeneous setting, different agents have different weights for the edges in the graph. The motivation of this model is that, in a company or a department, workers many need to work in pairs. Thus, if workers are viewed as items, an agent would like to have a set of pairs of workers so that the collective contribution for each pair of workers (modeled by the weight of the edge) is large.

The authors study MMS and EF1 allocations, and the design of approximation algorithm for optimizing the social welfare. Notice that the approximation ratio is not defined in its typical way in a constrained optimization problem. In particular, the social welfare output by the algorithm is not compared with the optimal social welfare *subject to MMS/EF1*; instead, it is compared with the optimal social welfare without any fairness constraint. This is like the algorithm-design aspect of the “price of fairness”. I think this is also okay.

For the homogeneous setting, the authors provide a greedy-based algorithm that output a 1/8-MMS allocation. This is done by a combination of multiple techniques, including the greedy technique in the scheduling-like problems, the rounding of edge weights to integer powers of 2, and the treatment of the hard special case where the maximum-value bundle contains a single matching edge. The authors also present an algorithm that returns an EF1 allocation with social welfare at least about 2/3 of the optimal social welfare. For the heterogeneous setting, the authors show that MMS cannot be satisfied for any bounded approximation ratio, even for two agents with binary weight functions. The authors also show that there is a gap of 1/n between the optimal social welfare and the best social welfare subject to EF1. However, for EF1, the authors present some nice positive results for 1) binary edge-weights and 2) two agents. In both special cases, we can obtain an EF1 allocation with 1/3 approximation to the optimal social welfare. Most technical details of these results are deferred to the appendix.


**Questions:**

 For the result with 1/8-MMS, do you have any tight examples?

**Limitations:**

The authors have addressed this in the conclusion section, where a list of future directions is provided. I think this is satisfactory.

**Strengths And Weaknesses:**

Strength:
1. The results are relatively complete for both the homogeneous setting and the heterogeneous setting. For MMS, we know it is achievable (with a constant factor approximation) for the homogeneous setting, but the price of MMS is unbounded; we know it is not achievable in the heterogenous setting. For EF1, it is always achievable. The price of EF1 is low for the homogeneous setting, and it is high for the heterogeneous setting. The authors also have completed the results for the two special cases with 1) binary edge-weights and 2) two agents. We know MMS is not achievable even when both 1) and 2) hold, and price of EF1 is low if either 1) or 2) holds. I would say the results are quite complete.
2. The 1/8-approximation algorithm for MMS is a neat one, although it is not very technically complicated. I have not checked the details in the appendix, so I am uncertain about the technical novelty for the remaining results. Nevertheless, it seems to me that there are some technical novelty for this paper.

Weakness:
1. My main concern to this paper is its model. I feel like the problem studied is too specific and the model has limited applications. I could understand the motivation provided by the authors. However, I still believe there are not a lot of practical scenarios where workers have to work in *pairs*. Why not groups of three or four, or even larger?
2. The complexity results (e.g., hardness-of-approximation) are mostly absent from this paper. The authors provide many tight examples for the approximation algorithms designed in this paper. I think this paper could be improved by a lot if some lower-bound results can be proved. However, I do not view this as a major weakness of the paper.

Overall, I think this is a borderline paper for a strong conference like NeurIPS. I like the fact that the results form a relatively complete landscape, and I also like the techniques behind those algorithms. On the other hand, I am uncertain about the significance of the problem studied in the paper (as I have addressed in 1 of “weakness”). I can see the motivation the authors provided, and I am convinced that the model may have some applications. However, I still do not think it is significant enough.

---

> ### Author Response · Authors · 2022-08-02
> **Response to Reviewer UCR6**
>
> Question: My main concern to this paper is its model. I feel like the problem studied is too specific and the model has limited applications. I could understand the motivation provided by the authors. However, I still believe there are not a lot of practical scenarios where workers have to work in pairs. Why not groups of three or four, or even larger?
>
> Answer: We thank the reviewer for the question.
> To motivate our model, we can consider the following real-world example.
> This example is also included in the revised version.
>
> Peer Instruction (PI) has been shown to be an effective learning approach based on a project conducted at Harvard University, and one of the simplest ways to implement PI is to pair the students [Crouch C H et al. 2001].
> Consider the situation when we partition students to advisors, where the advisors will adopt PI for their assigned students.
> Note that the advisors may hold different perspectives on how to pair the students based on their own experience and expertise, and they want to maximize the efficiency of conducting PI in their own assigned students.
> How should we assign the students fairly to the advisors?
> How can we maximize the social welfare among all (approximately) fair assignments?
> Our results shed light on solving these two questions.
>
> [Crouch C H et al. 2001] Crouch C H , Mazur E. Peer Instruction: Ten years of experience and results[J]. American Journal of Physics, 2001, 69(9):970-977.
>
> Question: For the result with 1/8-MMS, do you have any tight examples?
>
> Answer: The $1/8$ might not be tight. The approximation ratio of $1/8$ consists of three multiplicative approximate factors of $1/2$:
>
>    1.  Rounding all edge weights to powers of 2.
>
>    2. The greedy partition (i.e. $u(M_n) \geq (1/2) u(M_1) \geq  (1/2) MMS$).
>
>    3. The value of MMS is halved when halving the largest weight in the graph. % (i.e., $MMS(I')=(1/2) MMS(I)$)
>
> Actually, we can prove that all these three factors are tight, but the overall approximation ratio of $1/8$ is not tight. This is because, for example, the instance after step (a) (i.e., rounding the weights to powers of 2) does not match the tight example for step (b).
> We conjecture that with a more involved analysis, our algorithm has better than $1/8$ approximation.
> In the revised version, we added a sentence in this regard.

---

### Meta-Review · Area_Chair_pgU6 · 2022-08-26

**Recommendation:** Reject
**Confidence:** Certain

**Metareview:**

Reviewers agreed that the model is new and interesting and the theoretical results are solid. The main criticisms are about the model: some reviewers felt that it is too specific and there is not enough motivation. Some reviewers liked the technical depth while others felt that it is not enough to compensate for the lack of motivation. Overall, reviewers felt that the paper in its current form is not ready for publication at NeurIPS.

**Award:**

No

---

### Decision · Program_Chairs · 2022-09-14

Reject